# Distinction between pristine and disorder-perturbed charge density waves in ZrTe₃

Li Yue [1,6], Shangjie Xue [2,6], Jiarui Li [2,6], Wen Hu [3], Andi Barbour[3], Feipeng Zheng[4], Lichen Wang[1], Ji Feng [1,5], Stuart B. Wilkins [3], Claudio Mazzoli[3], Riccardo Comin [2]* & Yuan Li [1,5]*

Charge density waves (CDWs) in the cuprate high-temperature superconductors have evoked much interest, yet their typical short-range nature has raised questions regarding the role of disorder. Here we report a resonant X-ray diffraction study of ZrTe₃, a model CDW system, with focus on the influence of disorder. Near the CDW transition temperature, we observe two independent signals that arise concomitantly, only to become clearly separated in momentum while developing very different correlation lengths in the well-ordered state that is reached at a distinctly lower temperature. Anomalously slow dynamics of mesoscopic charge domains are further found near the transition temperature, in spite of the expected strong thermal fluctuations. Our observations signify the presence of distinct experimental fingerprints of pristine and disorder-perturbed CDWs. We discuss the latter also in the context of Friedel oscillations, which we argue might promote CDW formation via a self-amplifying process.

[1] International Center for Quantum Materials, School of Physics, Peking University, Beijing 100871, China. [2] Department of Physics, Massachusetts Institute of Technology, Cambridge, MA 02139, USA. [3] National Synchrotron Light Source II, Brookhaven National Laboratory, Upton, NY 11973, USA. [4] Siyuan Laboratory, Guangzhou Key Laboratory of Vacuum Coating Technologies and New Energy Materials, Department of Physics, Jinan University, Guangzhou 510632, China. [5] Collaborative Innovation Center of Quantum Matter, Beijing 100871, China. [6] These authors contributed equally: Li Yue, Shangjie Xue, Jiarui Li *email: rcomin@mit.edu; yuan.li@pku.edu.cn

Charge density waves (CDWs) and related phenomena have been a long-standing topic in condensed matter research[1–3]. A renewed interest in this topic was brought about in recent years by the discovery of ubiquitous signatures of CDWs in cuprate high-temperature superconductors[4,5]. On the one hand, the formation of CDWs in at least some of the cuprates is accompanied by sharp phonon dispersion (Kohn) anomalies[6], which are commonly found in conventional CDW systems. On the other hand, even though long-range CDWs can be stabilized in the cuprates by a variety of external fields[7–9], the three-dimensional (3D) ordering propagation vector is at odds with that of the Kohn anomalies in the zero-field condition[6,9]. In contrast, the zero-field charge correlations are often found to be short ranged[10–13] and coexisting with a rather inhomogeneous electronic background[13–17]. It is, therefore, of primary interest to elucidate the role of disorder during the incipience of the CDWs. In fact, it has been proposed that band structure effects[18–21], namely Friedel oscillations (FOs) seeded by impurities and quenched disorder, could produce experimental signatures that look similar to genuine CDWs[17,19,20,22]. Establishing an experimental methodology to characterize CDWs under the influence of disorder and elucidate their possible relation to FOs is hence important, but this is largely an outstanding task even in prototypical CDW materials.

A main challenge in experimentally addressing the role of disorder pertains to the detection length scale. Disorder is known to provide a local potential to foster the stabilization of dynamical charge correlations near the transition point[1,2,23–26]. How such nucleation actually occurs, and to what extent it promotes CDW formation, are questions that need to be addressed on top of the influence of disorder on the CDW correlation length and dynamics. They are all expected to occur at the mesoscopic scale, i.e., comparable to the CDW domain sizes that are often much greater than the density-modulation periodicity. Unfortunately, the majority of experiments carried out to date for addressing disorder effects in CDW materials fall either into the macroscopic regime, such as transport and thermodynamic measurements[27–29], or into the atomic-scale regime, such as scanning probe experiments[30,31].

For scattering experiments to survey the mesoscopic scale, special techniques are usually required. Here, we use resonant soft X-ray diffraction to study a prototypical CDW material, ZrTe$_3$, with very high accuracy. The length scale challenge is met on two fronts: first, we use soft X-rays in conjunction with a high-resolution area detector to achieve high momentum resolution, enabling us to distinguish Fourier signatures that are only slightly different. Second, we use a highly coherent X-ray beam to detect the domain texture and dynamics via interference patterns known as speckles[25,32,33]. The experiment further benefits from resonant enhancement of diffraction signals at the Te $M$ absorption edges —providing increased sensitivity to the very weak charge correlations near the CDW melting point even for relatively short acquisition times. These experimental advantages enable us to clarify the effects of disorder on the incipience of CDWs in ZrTe$_3$.

## Results

**System.** ZrTe$_3$ is a quasi-one-dimensional (1D) metal belonging to the monoclinic space group $P2_1/m$ (see Fig. 1a), with $a = 5.89$ Å, $b = 3.93$ Å, $c = 10.09$ Å, $\alpha = \gamma = 90°$, and $\beta = 97.8°$[34,35]. A pronounced resistivity anomaly can be observed at $63 \pm 1$ K (Supplementary Fig. 1), a temperature that is usually referred to as the CDW ordering temperature, $T_{CDW}$, in earlier literature[34–39]. The Fermi surface, contributed by four bands, comprises two sectors (Fig. 1b–d): 3D pockets around the Brillouin zone (BZ) center $\Gamma$, and quasi-1D sheets running along the BZ boundary with Fermi velocity primarily in the $\mathbf{a}^*$ direction[35,36].

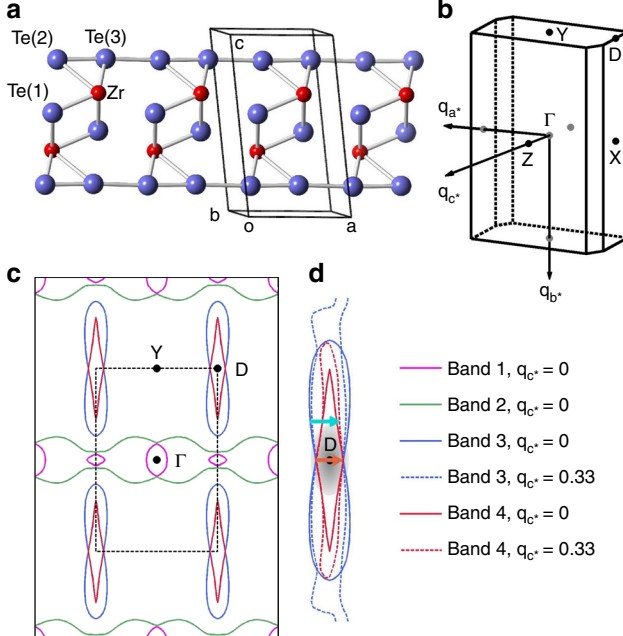

**Fig. 1 Crystal structure and Fermi surface of ZrTe$_3$. a** The crystal structure. **b** The first Brillouin zone. **c** Calculated Fermi surface (see Methods) at $\mathbf{q}_{c^*} = 0$. **d** Illustration of Fermi surface nesting properties. The Fermi surfaces indicated in red and blue are relatively flat sheets that run along the $\mathbf{c}^*$ direction perpendicular to the plane of display. Solid and dashed lines in **d** refer to the Fermi surfaces at $\mathbf{q}_{c^*} = 0$ and $\mathbf{q}_{c^*} = 0.33$, respectively, where the difference indicates Fermi surface warping along $\mathbf{c}^*$. The electronic gap associated with the CDW order opens from the shaded region around the D-point. The orange and cyan arrows indicate slightly different nesting wave vectors along $\mathbf{a}^*$ (with fixed 0.33 r.l.u. along $\mathbf{c}^*$) on different parts of the Fermi surfaces.

The CDW order is related to the nesting of the quasi-1D sheets that involve the $5p$ bands of Te(2) and Te(3) (Fig. 1a), with wave vector $\mathbf{q}_{CDW} = (0.07\mathbf{a}^*, 0, 0.33\mathbf{c}^*)$. The electronic gap associated with the CDW order opens near the D-point of the BZ[36,39], where electron–phonon coupling is strongest[34].

**Coexistence of two distinct signals.** In Fig. 2, we present temperature evolution of resonant X-ray diffraction signals near a $\mathbf{q}_{CDW}$ satellite of the $(0, 0, 1)$ fundamental Bragg reflection. Data at each temperature were acquired by performing a reciprocal space scan along the $\mathbf{c}^*$ direction, which involved rocking the sample while simultaneously repositioning the fast charge-coupled device (FCCD) camera in coupled fine steps, followed by reconstruction of the FCCD images into $(H, K, L)$ volume data (Fig. 2c). To highlight the diffraction signals, the FCCD's electronic background and fluorescence background (the latter due to the resonant probing conditions) have been measured separately and subtracted from the data. Thermal expansion of the lattice parameters has been accounted for based on separate non-resonant measurements of fundamental Bragg peaks in the same temperature range. As the mirror symmetry with respect to the $ac$ plane remains intact through the CDW transition, in Fig. 2a we visualize the $T$ evolution using a thin $(H, L)$ slice taken near $K = 0$. We make a few observations here:

First, a clear signal persists up to at least 64 K > $T_{CDW}$ (Fig. 2a). At even higher $T$, the diffraction pattern becomes smeared and weak to the point of being buried under the fluorescence background. Nonetheless, a CDW-related electronic gap[34,36] and

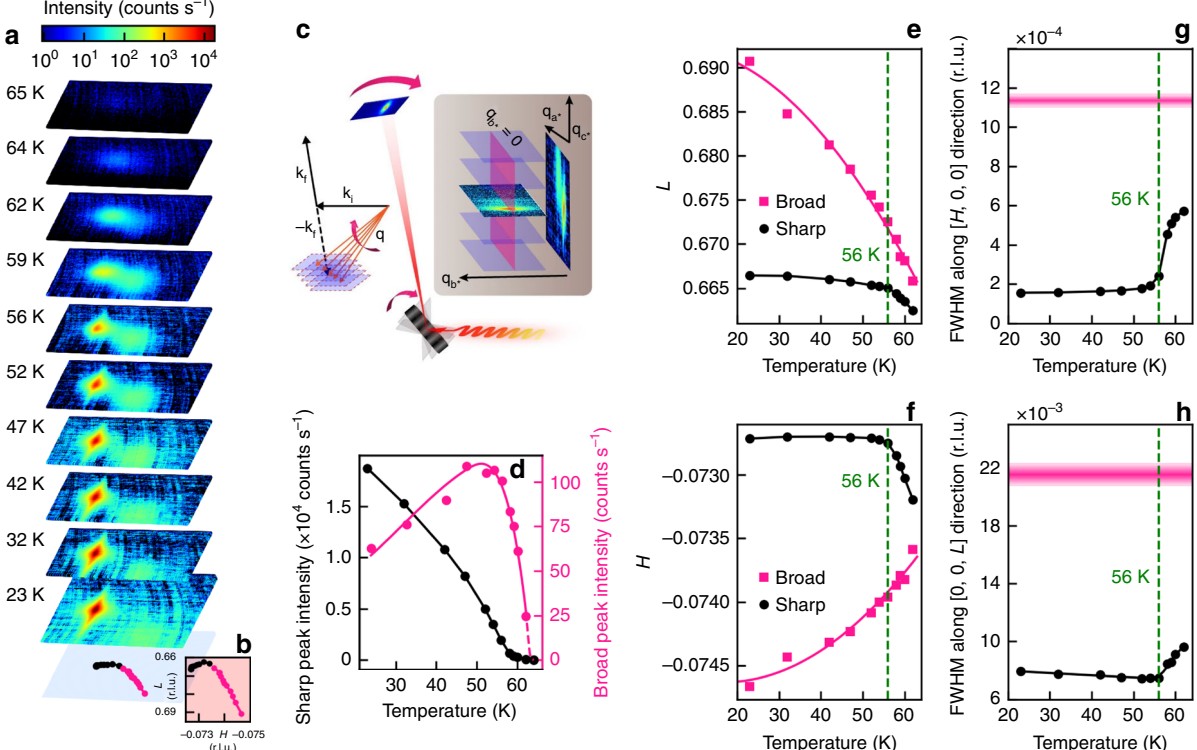

**Fig. 2 Two coexisting signals in reciprocal space. a** Reconstructed diffraction signals (see text and **c**) in the $[H, 0, L]$ plane at various temperatures. Each image's field of view is sufficiently large (note the larger field of view of the 23 K data set) to include the maxima of both peaks, enabling reliable extraction of their intensities. Ring-like patterns are due to incomplete background subtraction, and not related to presence of any impurity phases in our sample. **b** Momentum-space trajectory of the two peak centers indicated at the bottom of **a**. **c** Schematics of momentum scan (see text) in real and reciprocal space. The inset displays how the $(H, K, L)$ volume data were reconstructed and the $(H, 0, L)$ slice extracted. **d** Temperature dependence of the intensities of the sharp and broad peaks, determined from two-dimensional fits to the data (Supplementary Fig. 2). Lines are guide to the eye. **e**, **f** Temperature evolution of the momentum centroids of the broad and sharp peaks, projected along $L$ and $H$ directions, respectively. A leveling off is seen below ~56 K for the sharp peak. While the relative momentum changes can be determined quite precisely, there is a systematic uncertainty in the absolute readings of $L$ and $H$ caused by imperfect determination of the sample's orientation matrix, which we estimate to be 0.002 and 0.001 r.l.u., respectively. **g**, **h** Temperature evolution of the full width at half maximum (FWHM) of the sharp peak in $L$ and $H$, respectively. Horizontal shaded stripes indicate the FWHM (and its uncertainty) of the broad peak, which is weakly $T$ dependent.

suppression of phonon-damping effects[34,38] have been observed up to higher temperatures ≈ 140 K, indicative of persisting short-range charge correlations and fluctuations above $T_{CDW}$. Diffuse scattering signals centered around $\mathbf{q}_{CDW}$, presumably related to thermally activated soft phonons, have been observed in hard X-ray scattering experiments even up to room temperature[37].

Second, two coexisting peaks materialize almost immediately below $T_{CDW}$. While both peaks are centered at $K = 0$ independent of $T$, they are gradually separated in the $(H, L)$ plane upon further cooling (Fig. 2b, e, f), and develop very different momentum widths at low $T$ (Fig. 2a, g, h, note the logarithmic color scale in Fig. 2a). Given the distinct $\mathbf{q}$ widths, hereafter we refer to them as the broad and the sharp peaks.

Third, we identify a new characteristic temperature, $T_{LO} = 56 \pm 3$ K. Below $T_{LO}$, the sharp peak's $\mathbf{q}$ position becomes nearly insensitive to $T$ (Fig. 2e, f), and its $\mathbf{q}$ widths become narrow and saturated (Fig. 2g, h and Supplementary Fig. 3), amounting to minimal correlation lengths of ~9000, 1000, and 400 Å along $\mathbf{a}^*$, $\mathbf{b}^*$, and $\mathbf{c}^*$, respectively. These indicate that a truly long-range CDW order forms and the associated charge modulation periodicity reaches a stationary stage below $T_{LO}$. In contrast, the broad peak continues to evolve in its $\mathbf{q}$ position, and remains broad (broader than the sharp peak by a factor of 3–6) below $T_{LO}$.

Finally, it is also below $T_{LO}$ that the sharp peak gains the majority of its intensity (Fig. 2d, and Supplementary Fig. 4),

rather than below $T_{CDW}$. The broad peak, in contrast, increases rapidly below $T_{CDW}$, reaches its maximum at about $T_{LO}$, and then gradually decreases upon further cooling (Fig. 2d).

The presence of two distinct signals near $\mathbf{q}_{CDW}$ is a surprising yet robust result. The two signals have been observed in two different samples that produced sufficiently narrow and strong peaks, and the intensity of the broad peak varied between the two samples. This hints at a disorder-related origin of the broad peak, in which case sample-specific inhomogeneity can explain the intensity variation. We emphasize that only high-quality samples allowed us to separate the two signals—measurement on a purposely quenched and disordered crystal yielded only one much broader peak (Supplementary Fig. 5) at all temperatures below $T_{CDW}$, precluding the identification of the two signals.

**Physical origin of the two signals.** In a conventional diffraction experiment (e.g., using a point detector), one would mainly notice the sharp peak since it dominates the total scattering intensity. It is therefore most natural to attribute the sharp peak to the CDW order parameter. We note, however, that in a previous hard X-ray diffraction experiment[37], the CDW superstructure intensity was found to rise rapidly below $T_{CDW}$, rather than below $T_{LO}$, which was not seen in the previous study. Even though we find that the broad peak's intensity does rise rapidly below $T_{CDW}$ (Fig. 2d), given the greater intensity of the sharp peak (Supplementary Fig.

4), a simple summation of both peaks in our case would not reproduce the previous hard X-ray result.

We believe that the discrepancy is caused by different scattering cross sections involved in our resonant and the previous nonresonant experiments. Hard X-ray diffraction is mainly sensitive to lattice displacements, whereas resonant (soft) X-ray diffraction is mainly sensitive to density modulations of conduction electrons. The combination of the previous[37] and our results is particularly revealing, and it point toward a two-step linkage of lattice deformations to charge modulations. Upon cooling at first, i.e., for $T_{LO} < T < T_{CDW}$, the rapidly deforming lattice[37] is accompanied by short-range charge modulations that give rise to the broad peak (whereas the sharp peak is still in its incipience, see discussions later); then, for $T < T_{LO}$, prominent and long-range charge modulations finally form, which bear the true nature of CDWs. Importantly, the second step takes place at the cost of suppressing the broad peak.

As discussed toward the end of the previous subsection and supported by further evidence in the next subsection, the broad peak's signal ought to come from regions in our sample that contain disorder. With alternative possibilities to be discussed later, here we consider it most likely that the signal arises from standing waves created by self-interfering electrons scattered off a localized potential, i.e., FOs, or is closely related to FOs. This interpretation is consistent with the fact that the broad peak is always broad even in the CDW-well-ordered state, as FOs are short-ranged per se. It also explains why the broad and sharp peaks initially emerge together in **q**-space, as FOs and CDWs are linked to the same Fermi surface nesting instability at $T > T_{CDW}$. The departure of the two wave vectors from each other upon cooling below $T_{CDW}$ can be understood by appreciating the **k**-dependent property of the nesting. Denoting pair-wise nesting as involving electronic states at momenta **k** and $\mathbf{k} + \mathbf{q}(\mathbf{k})$, both of which are restricted to the Fermi surfaces and $\mathbf{q}(\mathbf{k})$ depends on **k**, the overall nesting wave vector $\mathbf{q}_{avg}$ is determined by the (weighted) average of $\mathbf{q}(\mathbf{k})$ over **k**. At $T = T_{CDW}$, $\mathbf{q}_{avg}$ ($=\mathbf{q}_{CDW}$) is primarily contributed by electronic states with **k** close to the D-point of the BZ Fig. 1c), where strong electron–phonon coupling puts high weights in this region in the **k**-average for the purpose of CDW formation[34]. Indeed, the CDW electronic gap is most prominent in this region[36,39]. As these electronic states are removed from the Fermi surface by the gap, however, the remaining states available for nesting may require a slightly different $\mathbf{q}_{avg}$ than $\mathbf{q}_{CDW}$, making the FO and the CDW wave vectors depart (depicted by cyan and orange arrows in Fig. 1d, respectively). In practice, this qualitative picture must be supplemented by quantitative details, including the $T$ and the **k** dependence of the CDW gap and the electron–phonon coupling matrix elements, in order to produce results that can be compared with our experiment. Yet the most important merit of this scenario, namely its natural explanation of the broad peak's same momentum as the sharp peak at $T_{CDW}$, and its intensity decrease below $T_{LO}$ (Fig. 2d) as due to a depletion of the best-nested part of the Fermi surfaces caused by the CDWs, is independent of the details.

The abrupt increase of the broad peak below $T_{CDW}$, in conjunction with the previously reported[37] lattice deformations below the same temperature, points to a positive feedback mechanism between the coupled lattice and electronic degrees of freedom. Under our interpretation of the broad peak as arising from FOs, it follows that upon cooling in the range $T_{LO} < T < T_{CDW}$, the lattice rapidly deforms around impurities and defects due to the local presence of FOs, which in turn enhances the disorder scattering potential and gives rise to stronger FOs. The strong momentum (**k** and **q**) dependence of

such feedback mechanism[34] may explain why the FO signals materialize in the form of a reciprocal-space peak, rather than a contour[17], as Kohn anomalies that require electron–phonon coupling also commonly exist only in highly restricted momentum regions[6,37]. Below $T_{LO}$, long-range CDWs form on top of the deformed lattice, which then suppresses the disorder scattering and lead to a decrease of the broad peak. Importantly, transport data also support our interpretation by showing a semiquantitative correspondence between resistivity variation (due to enhanced scattering) and the broad peak's intensity below $T_{CDW}$ (Supplementary Fig. 6). In this regard, we argue that $T_{CDW}$ merely marks the onset of the positive feedback between the lattice and the FOs, and the true long-range CDWs form only at $T_{LO}$, below which the resistivity (along $a$-axis) actually decreases again.

Related to our above observations and interpretation, non-resonant X-ray diffraction signals associated with FOs have previously been reported in the quasi-1D CDW material $K_{0.3}$($Mo_{0.972}V_{0.028}$)$O_3$ with controlled disorder introduced through vanadium impurities[40]. In light of our results, the previously observed asymmetric profile of the CDW satellite reflections, similar to our data at 62 K (Supplementary Fig. 2), may indeed originate from a sum of two (CDWs and FOs) signals[40], but the high impurity concentration of the previous sample did not allow the authors to observe a clear and progressive separation of the signals, nor to address their relation on the verge of CDW formation. Being able to observe such progressive separation but without noticeable change in the width of the broad peak with cooling, we further demonstrate that the separation we observe (Fig. 2) is unrelated to the (short) correlation length of the FOs, which was used to explain the previously observed separation[40]. In a more recent study of heavy-Fermion compounds[41], Gyenis et al. demonstrated that resonant X-ray diffraction indeed seems to possess the sensitivity to detect FOs, the signal of which exhibited resonant enhancement and increased toward low $T$ similar to our results. However, the absence of CDW order in those materials precluded a differentiation of CDW and FO signals, a methodology that is much needed in our context.

To end this subsection, we emphasize that our direct observation of coexisting sharp and broad diffraction signals, along with their distinct temperature evolution, is new and revealing. They are the signatures of pristine and disorder-induced charge correlations, respectively. Hereafter, we present another method to highlight the influence of disorder by investigating the mesoscopic dynamics, which must be different between spontaneous and disordered-induced effects.

**Mesoscopic dynamics.** Figure 3a, b displays a comparison of FCCD images of the CDW diffraction signals taken before and after the insertion of a 10 $\mu$m pinhole into the X-ray beam's path ~5 mm upstream from the sample. With the pinhole, which improved the beam coherence and reduced the beam size, we observed speckle patterns (Fig. 3b) due to the interference of the X-rays diffracted from different charge domains. The beamline's high photon flux allowed us to record a statistically significant speckle pattern within less than a minute of photon accumulation, and its high stability allowed us to monitor the pattern over a time span up to hours. Dynamical charge domains manifest themselves as time-varying speckle patterns. In Fig. 3c–f, we present waterfall plots of the speckle pattern time series recorded at several temperatures, which are constructed by vertically stacking narrow horizontal sections (11-pixel wide, see dashed line in Fig. 3b) of FCCD data acquired at different times. The continuous and straight vertical streaks in Fig. 3c, f indicate that the speckle patterns are very stable at 23 and 62 K, whereas the

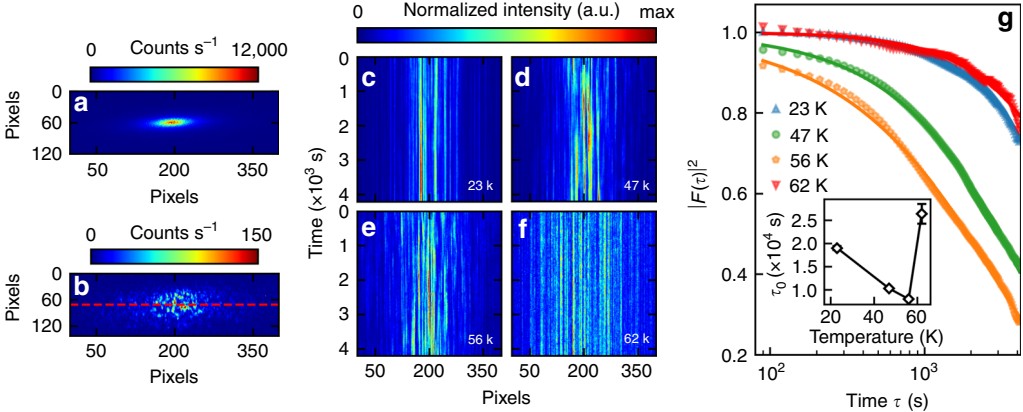

**Fig. 3 Mesoscopic dynamics of charge domains. a, b** Images of CDW diffraction signal taken at 45 K without (**a**) and with (**b**) the 10 $\mu$m pinhole (see text). Due to the much lower count rate in **b**, the image is obtained from an average of three frames, each with 10 s of exposure time. **c–f** Waterfall plots (see text) of the time series of diffraction intensities extracted from a horizontal strip taken near the dashed line in **b**. Global intensity variations over time (Supplementary Fig. 7) have been compensated by normalizing the mean intensity of each time frame. **g** Autocorrelation of speckle patterns at different temperatures. The inset shows fitted values of the coherence time $\tau_0$ (see text) that characterizes the rate of domain motion. Error bars in the inset represent uncertainties (1 s.d.) obtained from least-square fitting of the data. The autocorrelation at 62 K may exhibit some initial rapid decay (Supplementary Fig. 8) that is not accounted for by the fit.

broken streaks in Fig. 3d, e suggest presence of mobile domains at these intermediate temperatures. The same conclusion can be captured more quantitatively by analyzing the intensity autocorrelation function[32] of the speckle patterns:

$$g_2(\tau) \equiv \frac{\langle I(p,t)I(p,t+\tau)\rangle}{\langle I(p,t)\rangle^2} \equiv 1 + \beta|F(\tau)|^2, \qquad (1)$$

where $I(p,t)$ is the intensity obtained at time $t$ and pixel $p$, $\tau$ is the time difference, and $\beta$ is a measure of the contrast of the speckle patterns. The average $\langle\cdots\rangle$ is taken over $t$ and all pixels in a small region near $\mathbf{q}_{CDW}$. $F(\tau)$ is the intermediate scattering function, for which we assume an exponential form:

$$|F(\tau)| = e^{-(\tau/\tau_0)^\alpha}, \qquad (2)$$

where $\tau_0$ represents the characteristic time required for the domain distribution to change significantly, and $\alpha$ is the stretching exponent[32], determined to be around unity in our experiment (Supplementary Fig. 8). It can be seen that the dynamics are faster at 47 and 56 K than at 23 K, which can be explained by thermal activation of CDW domain walls. However, upon further heating to 62 K, which is close to or even slightly above the melting point of the CDWs, the dynamics become slow again.

The anomalously slow dynamics at 62 K are very difficult to explain in terms of regular CDW domain motion. They are, however, consistent with the notion that the system's evolution between $T_{LO}$ and $T_{CDW}$ mainly takes place around disorder sites. The broad diffraction signal (Fig. 2), which we attribute to FOs, should indeed always produce static speckles since defects do not move at cryogenic temperatures. However, the broad and sharp peaks partially overlap at 62 K (Fig. 2a and Supplementary Fig. 2), and it is the sharp peak that dominates the total intensity, so the static speckles are most likely contributed by the sharp peak. In the following, we discuss how the sharp peak may produce dynamic speckles at 47 and 56 K yet static ones at 23 and 62 K, as well as the formation process of the CDWs newly revealed by our results altogether.

## Discussion

Figure 4 schematically illustrates the evolution of charge modulations with temperature in the dilute-disorder limit, which we believe applies to our high-quality ZrTe$_3$ samples. In

this limit, incipient CDWs are stabilized and pinned around disorder sites up to a certain distance, and there is an appreciable temperature range above the formation temperature of long-range CDWs ($T_{LO}$), where the charge modulations are sufficiently strong to produce a resonant diffraction signal with coherent speckles. But as the correlation length is still short, the modulation domains are disconnected from one another (Fig. 4a). Due to the pinning, these isolated domains are highly static, hence explaining the static speckles observed at 62 K. This phenomenon is similar in essence to the anomalous static central peak observed above structural phase transition temperature in SrTiO$_3$[42]. Upon further cooling toward and through $T_{LO}$, the correlation length rapidly increases so that the disconnected domains merge into long-range CDWs, and it is at this point that regular CDW domain walls form (Fig. 4b). As long as the domain walls are not too rigid to be thermally perturbed, domain redistribution is expected to occur over time, consistent with our observations at 47 and 56 K in Fig. 3. Finally, at very low temperatures (23 K in Fig. 3), the CDW fabric becomes stiff, and its domain walls are robust against thermal perturbations (Fig. 4c).

The nucleation and pinning mechanism between $T_{LO}$ and $T_{CDW}$ deserves a separate note. According to our analysis above, the speckles observed at 62 K are mainly contributed by the sharp peak, not by the broad peak as per consideration of their intensity comparison. Nevertheless, we find corroborating evidence that the broad peak becomes strong via self-amplification enabled by a lattice–electron positive feedback loop between $T_{LO}$ and $T_{CDW}$. Under our interpretation that the broad peak arises from FOs, these FOs locally have the same (broken) translational symmetry as the CDWs, so they serve as a conjugate field that induces a crossover behavior of the CDWs above the genuine transition temperature, which we believe is $T_{LO}$. The induced CDWs are pinned and as static as the FOs, which explains the observed static domains at 62 K. Similar in essence to our picture, incipient and incoherent CDW nanopatches have been observed in other CDW systems at temperatures and intercalation conditions[43–45] beyond the vanishing point of bulk long-range order, and our study here helps to elucidate the role of a deformable lattice in giving rise to such phenomena. Our interpretation does not require a direct pinning of domain walls, since in the dilute-disorder limit, all domain walls are expected to end up in pristine regions of the crystal (Fig. 4c).

**Fig. 4 Schematics of charge modulations near disorder in different temperature regimes. a** $T_{LO} < T < T_{CDW}$ (62 K in Fig. 3), **b** $T \lesssim T_{LO}$ (47 and 56 K in Fig. 3), and **c** $T \ll T_{LO}$ (23 K in Fig. 3). Black dots represent disorder. The red modulations correspond to the broad peak in Fig. 2, whereas the grey modulations correspond to the sharp peak. Each panel contains a superposition of both modulations, with the grey ones (CDWs) progressively increasing toward low $T$, first in correlation length (from **a** to **b**), then in intensity (from **b** to **c**). In **b** and **c**, we add an exaggerated 10% difference between the periodicity of the two patterns, making the maximal intensities of the two modulations shift from each other (arrows in dashed circle in **c**), in order to reflect our experimental observations in Fig. 2.

Indeed, if the FOs and the induced CDWs have identical (broken) symmetry, they must be linearly coupled and any conceptual boarder line between them must be blurred. But as can be seen from Fig. 2d, the induced CDW crossover is rather weak in the sense that the majority of the sharp peak's intensity forms below $T_{LO}$. The weak crossover is probably related to the slight detuning of the FOs' spatial periodicity from that of the CDWs (Fig. 2a). Microscopically, the detuning may arise from a competition between the FOs and the (induced) CDWs for the same Fermi surfaces, as we have discussed earlier. The competition continues below $T_{LO}$: The CDWs finally win, suppress the FOs' magnitudes, and keep pushing the FOs away in **q** with cooling (see Fig. 2e, f and illustration in Fig. 4c).

There are alternative, albeit in our opinion more remote, ways to understand the origin of the broad peak and its momentum detuning from the sharp peak. One possibility is that it arises from directly induced CDWs in the immediate neighborhood of disorder sites. In contrast, in both this scenario and our original interpretation based on FOs, the sharp peak at $T_{LO} < T < T_{CDW}$ arises from CDWs that are indirectly induced and pinned at distance from the disorder, since it dominates the signal and produces the static speckles, and because it continuously evolves below $T_{LO}$ into the peak of truly long-range CDWs. Regarding the detuning in **q** between the two peaks, although the presence of impurity in ZrTe$_3$ may locally modify the electronic structure[39] or create topological defects in the CDWs, it remains to be investigated why the detuning effect gradually develops below $T_{CDW}$ and is not present at $T_{CDW}$.

Another possibility is that the broad peak arises from thermal diffuse scattering, since soft phonons (i.e., the Kohn anomaly[37]) are known to prevail at $\mathbf{q}_{CDW}$ near the transition. However, the small momentum transfer of soft X-ray scattering makes it not particularly sensitive to phonons. Even if strong electron–phonon coupling may allow us to see the Kohn-anomalous phonons, the diffuse scattering intensity is expected to reach its maximum at $T_{CDW}$, where the phonons are softest[37], and the sharp and broad peaks are about to form at the same $\mathbf{q}_{CDW}$, rather than near $T_{LO}$, where the two peaks are well separated. Therefore, while thermal diffuse scattering might be the origin of the weak signals that persist above $T_{CDW}$ (Fig. 2a), we rule out the possibility that it produces the broad peak seen in the range $T_{LO} < T_{CDW}$.

Our results are consistent with the notion that the formation of long-range CDWs in realistic materials is seeded by disorder[1,2,23–26]. Yet beyond conventional wisdom, we establish the presence of two distinct temperatures, $T_{LO}$ and $T_{CDW}$, in a prototypical CDW system. Their existence allows for an experimental differentiation of scattering signals caused by CDWs that are pristine and strongly perturbed by disorder: The latter may receive substantial promotion from FOs, which are always present

in metals. We believe that in materials with sufficiently strong CDW tendency, which boils down to good Fermi-surface nesting and strong electron–phonon coupling, FOs have the opportunity to grow in a self-amplifying fashion due to the help of lattice soft-mode condensation below $T_{CDW}$, and it is these enhanced FOs that stabilize a density modulation of the otherwise uniform electron liquid at distance from disorder. Such CDWs, induced indirectly rather than directly by disorder, give rise to the sharp peak that is distinct from the broad peak which comes from the immediate neighborhood of disorder, and together they form disconnected and static charge domains at $T_{LO} < T < T_{CDW}$. $T_{LO}$ can thus be regarded as the purely electronic CDW ordering temperature, around which the domains spontaneously merge together, making mobile charge domain walls most noticeable.

In light of our results, it is interesting to notice that when long-range CDWs form in the cuprate superconductors[7–9], the sharp diffraction peak is accompanied by a much broader peak associated with short-range CDWs. While this coexistence of two signals bears some similarity to our finding in ZrTe$_3$ below $T_{CDW}$, the two systems are very different both in the signals' amount of separation in **q**, and in their dependence on temperature (and external fields). Hence, we do not think a direct analogy between the CDWs (and the FOs) in ZrTe$_3$ and the cuprates should be drawn at this point. Yet empirically, it does seem that even the rather long-range CDWs studied recently in La$_{1.875}$Ba$_{0.125}$CuO$_4$ lack the phenomena that we have seen: only one peak (considerably broader than ours) and one characteristic temperature were observed, and the speckle patterns were found to be always static at low temperatures[33] and with memory effects up to high temperatures[46]. The counterpart of our finding, namely, the role of disorder in charge-order formation in the strong-coupling limit, to which the cuprates belong, deserves further experimental scrutiny.

## Methods

**Calculation of the Fermi surface.** Each cut of the Fermi surface on $\mathbf{a}^*\mathbf{b}^*$ plane was calculated using Quantum Espresso package[47] based on density-functional theory, within the generalized-gradient approximation parameterized by Perdew, Burke, and Ernzerhof[48,49]. Norm-conserving pseudopotentials, generated by the method of Goedecker, Hartwigsen, Hutter, and Teter[50], were used to model the interactions between valence electrons and ionic cores of both Zr and Te atoms. The Kohn–Sham valence states were expanded in the plane-wave basis set with a kinetic energy truncation at 150 Ry. The equilibrium crystal structure was determined by a conjugated-gradient relaxation, until the Hellmann–Feynman force on each atom was less than $0.8 \times 10^{-4}$eV Å$^{-1}$ and zero-stress tensor was obtained. A $12 \times 18 \times 8$ **k**-grid centered at the Γ point was chosen in the self-consistent calculation, following by a non-self-consistent calculations on a **k**-grid of $54 \times 78 \times 1$ to obtain the Fermi surfaces. A Gaussian-type broadening of 0.0055 Ry was adapted.

**Scattering experiment.** Single crystals of ZrTe$_3$ were synthesized by a vapor transport method[34]. Resonant soft X-ray scattering experiments were performed at

the NSLS-II facility (Brookhaven National Laboratory) on beamline 23-ID-1, which provides coherent X-rays with a high flux of $\sim 10^{13}$ photons per second and excellent mechanical stability. A FCCD camera with a maximal readout rate of 100 Hz and $30 \times 30 \, \mu m^2$ pixel size was placed 34 cm away from the sample, which was mounted with the $[H, 0, L]$ reciprocal plane lying in the vertical scattering plane. The measured CDW signals were located at about $(-0.07, 0, 0.67)$ in reciprocal lattice units (r.l.u.), a satellite reflection near the $(0, 0, 1)$ Bragg peak. The horizontally polarized incident beam was tuned to a photon energy of 630 eV, in order to maximize the CDW diffraction signal (Supplementary Fig. 9).

**Fitting of the diffraction signals**. In order to extract the temperature-dependent parameters that describe the broad and sharp peaks in Fig. 2, we use two two-dimensional Gaussian profiles with anisotropic widths. The results are displayed in Fig. 2 and Supplementary Figs. 2–4. A detailed study of the line-shape characteristics of the data, as has been done for instance in ref. [51], is beyond the scope of our present study.

## Data availability

The data that support the plots within this paper and other findings of this study are available from the corresponding authors upon reasonable request.

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

## Acknowledgements

We wish to thank M. Dean, B. Fine, M.-H. Julien, B. Keimer, S. A. Kivelson, M. Le Tacon, M. Minola, Y. Y. Peng, N.-L. Wang, Yan Zhang, and Yi Zhang for discussions. The work at PKU is supported by the National Natural Science Foundation of China (NSFC) under

grant no. 11888101 (Y.L.), by the National Basic Research Program of China under grant nos. 2018YFA0305600 (Y.L. and J.F.) and 2015CB921302 (Y.L.), and by the NSFC under grant no. 11725415 (J.F.). The work at MIT is supported by the National Science Foundation under grant no. 1751739. The work at Jinan University is supported by the NSFC under grant no. 11804118. This research used beamline 23-ID-1 of the National Synchrotron Light Source II, a U.S. Department of Energy (DOE) Office of Science User Facility operated for the DOE Office of Science by Brookhaven National Laboratory under contract no. DE-SC0012704. Additional measurements were performed at the REIXS beamline of the Canadian Light Source during the review process, with the assistance of Wenjie Chen, Zach Anderson, Ronny Sutarto, and Feizhou He. Part of the calculation was supported by High Performance Computing Cluster in Jinan University.

## Author contributions

Y.L. and R.C. conceived the research. L.Y. and S.X. prepared the samples. L.Y., S.X., J.L., W.H., A.B., L.W., S.B.W., C.M., R.C. and Y.L. performed the X-ray scattering experiments. L.Y., S.X., J.L., C.M., R.C. and Y.L. analyzed the data and interpreted the results. F.Z. and J.F. performed the first-principles calculations. L.Y. and Y.L. wrote the manuscript with input from all coauthors.

## Competing Interests

The authors declare no competing interests.
