## [Peer Review File · Nature Communications]

Reviewers' comments:

Reviewer #1 (Remarks to the Author):

The influence of defects on the formation of CDWs is a very interesting topic and still little studied by modern 3rd generation instruments, especially when fluctuations close to the phase transition are taken into account. For this reason, this article deserves all the attention.

Having said that, there are a number of issues, both formal and substantive, that are prohibitive for any publication.

The subject of Freidel Oscillations is particular, little studied by diffraction (a single publication cited in the paper to my knowledge), not easy to observe by diffraction and whose characteristics generate a very original X-ray signature (an asymmetrical contribution in the feet of the 2kF satellite). Evoking the presence of Freidel oscillations to explain the measured asymmetric diffraction pattern requires a much more precise presentation of the foundations of this approach. You must convince the reader that this asymmetrical contribution cannot come from anything else, especially in your case, where the crystal seems to be of poor quality (presence of many rings in the diffraction patterns in Figure 1).

The data presentation has also to be redone. Figure 1 is incomplete (missing the FWHM, to put in the main text and not in the supplement, it is an essential information) and not very understandable (the variations of the 2 wave vectors are unclear; the addition of a point on the maximum intensity of the diffuse peak is not recommended; is the diffuse peak comes out of the measuring window at low T leading to an artificial decrease of intensity?). There is also missing data concerning the diffuse intensity still present, according to you, above T_c up to 140K.

In general, the article gives an impression of superficiality because of missing data and a lack of precise explanation of the results (why does this diffuse peak drop in intensity below 50K? Why such temporal fluctuations? Do Bragg reflections show the same asymmetrical contribution - how behave the other satellites? why this broad peak moves in the opposite direction with temperature?), despite the obvious interest of the topic.

Reviewer #2 (Remarks to the Author):

This manuscript applies state-of-art resonant X-ray scattering (REXS) to a charge-density-wave (CDW) material, ZrTe₃, aiming to distinguish between different types of charge ordering. The key result is shown in Fig. 2(a), which describes the temperature dependence of the CDW diffraction peak. Surprisingly, it is found that the main peak is accompanied by a satellite peak at a close-by wave-vector. This interesting experimental fact had not been described or predicted before and deserves publication.

At the same time, I think that the manuscript contains many unjustified claims and conjectures and requires a complete rewriting:

1. The paper is framed in the context of Cuprate high-temperature superconductor, where REXS have become very popular thanks to the detection of a short-ranged CDW order by Ghiringhelli et al (Science 2012). Since then, hundreds of papers (including tens of Science and Nature papers) were written trying to establish whether the signal is due to a true competing CDW order, or merely, to Friedel oscillations (FO). The present manuscript claims that the double peak observed in ZrTe₃ helps clarifying this debate: They associate the main and satellite peaks, respectively, to CDW and to FO. This situation is apparently similar to the high-magnetic-field measurements of Gerber et al (Ref. [7]), where two distinct diffraction peaks were found. However, there are two main differences:
 - a. In Cuprates, the short-ranged FO and the long-ranged CDW appear at different temperatures and/or magnetic fields, while in ZrTe₃ they appear exactly at the same temperature.
 - b. In Cuprates, FO and CDW appear at well-separated wave-vectors (in the “c” axis component), while in ZrTe₃ they appear one next to the other.

Because of these two key differences, I do not think that the present study is directly relevant to Cuprates, nor that the two peaks are associated with FO and CDW oscillations.

2. The paper makes contradictory claims about the CDW transition temperature. These claims are based on linear extrapolations of the REXS signal and suggest transition temperatures ranging from 52K to 59K. This large uncertainty needs to be contrasted with the sharpness of the resistivity measurement, which shows a pronounced lambda peak at 63K. Because the latter experiment gives a much sharper signal, I do not think that the linear extrapolation performed by the authors has any scientific relevance. (Just for fun, I attach a plot in which a different linear extrapolation gives a transition temperature of 64K). Similarly, I do not understand why the critical temperature should correspond to a saturation of the FWHM width. This saturation implies that the CDW signal is full

developped: It always occurs much below the critical temeperature (where the long-range component of the CDW starts developping and is still accompanied by strong short-range fluctuations). Finally, the conjecture that the diffraction signal may be present already at 140K is not supported by the experimental findings and should be erased from the text.

3. As mentioned above, the experiment shows that the two peaks are born at the transition temperature of the CDW phase, $T=64\pm 1\text{K}$. This finding is not consistent with the schematic plot of Figure 4, where the FO start developing at high temperatures and progressively become longer and longer ranged. Furthermore, the picture presented in Figure 4 does not explain the emergence of a satellite peak in the X-ray scattering at low temperature (which is the main experimental finding of this paper).
4. The authors claim that Ref. [38] was not able to separate CDW and FO oscillations. I do not agree with this statement. See in particular Fig. 1 there, where two peaks at wavevectors q_c and q_s are detected. I think that the authors should give more highlight to this earlier publication, which demonstrated for the first time that X-ray scattering can probe Friedel oscillations!
5. One possibility is that the satellite peak is due to FO induced by the domain walls of the CDW order (This may explain the non-monotonous temperature dependence). However, how can one explain the tiny momentum difference between the main CDW peak and the satellite peak? I find it hard to believe that this is due to the nesting between two parallel bands, along the lines of Ref. [19-21].

To summarize, I think that this scientific work is very timely and constitutes an important advancement of the field. REXS has recently become one of the leading probes to describe strongly correlated materials, and high-temperature superconductors in particular. Hence, it is very important to apply REXS to other known materials, to understand the limitations and advantages of this probe. This paper offers a much-needed independent test of this technique. However, the text of the manuscript has to undergo a major revision before being considered in Nature Communications.

Reviewer #3 (Remarks to the Author):

The paper by Yue et. al. represents an interesting study on the charge density wave (CDW) compound ZrTe_3 , showing that there are two contributions to the CDW superlattice reflection. One comes from the long-range order, which shows up below $\sim 55\text{K}$, while the other weaker and broader reflection, shows up at a different wavevector and at slightly higher temperature ($\sim 63\text{K}$). The authors interpret this secondary reflection as arising from "Friedel oscillations" seeded by impurities. The authors present supplementary evidence of their interpretation using intensity correlation measurements, showing time dependent signals only in the intermediate temperature range below the transition temperature. Overall, the paper is written well, and most points are made with clarity. If I understand the text correctly (which I am not sure that I do, see the comments below), the authors are likely correct in their interpretation of their data. Overall, the work is solid, the technical achievements in this work are noteworthy (the momentum resolution achieved and the coherent diffraction speckle), and the observations are interesting enough to warrant publication in Nature Communications if some major revisions are made.

1) My main point of confusion, which was alluded to above, concerns what the authors refer to as "Friedel oscillations". Naively, I imagine "Friedel oscillations" as something unassociated with a charge density wave but can arise in any Fermi gas/liquid. However, both Friedel oscillations and a charge density wave instability occur nominally at twice the Fermi wavevector. What I do not understand from the text is whether the authors are trying to draw a distinction between "Friedel oscillations" and an impurity-pinned short-ranged CDW or whether they are referring to them as the same phenomenon. My comments for either interpretation are presented below.

a. If the authors are trying to distinguish the two as different physical phenomena, a couple sentences need to be written in the introduction explaining how they are different. Furthermore, there would need to be strong evidence in the data that is able to differentiate between the two, which would need to be discussed in the text after the data is presented. Naively, I do not necessarily see the difference between the two in the data presented by the authors. Does resonant soft X-ray scattering help distinguish the two somehow in a way that I am unaware?

b. If the authors are referring to Friedel oscillations and the short-ranged CDW as the same physical phenomenon, they have chosen a rather unconventional terminology, which is likely to confuse other readers as well. In my reading, this is what I understood the paper to mean, and I think this interpretation of the data is likely to be correct. To those used to studying conventional charge density waves, one usually refers to the phenomenon as a short-ranged CDW or a localized impurity-pinned or defect-pinned CDW. Indeed, if the authors are equating the two physical phenomena, I would suggest that the authors change their terminology. Indeed, the STM paper by Arguello et. al. [PRB 89, 235115 (2014)] on NbSe_2 discusses at length why what they observe should be interpreted as a short-ranged CDW stabilized by defects rather than being equated with Friedel oscillations.

Other than this rather major issue in the paper, I have some other minor comments which the authors should address:

1) Nowhere in the main text do the authors talk about the length scales associated with either CDW. Since the authors consistently refer to the peaks as "broad" and "sharp", it would be beneficial to the reader to understand the length scales involved, even if just a lower bound on the length scale of the sharper peak is presented (which is likely limited by resolution). The data is ultimately presented in the supplement, but it would be helpful to have an estimate in the main text.

2) The authors may want to draw the nesting vector on the Fermi surface in Fig. 1(c). To those not familiar with the material, this would be beneficial.

3) Some plots refer to a peak amplitude and some refer to a peak intensity. I would use the latter terminology, as it is less ambiguous. Unless, of course, the authors do mean to draw a distinction, in which case I would suggest clarifying what is meant.

4) The authors do not cite several references which appear to be highly relevant to their observations.

The STM papers by Arguello et. al. [PRB 89, 235115 (2014)] and Chatterjee et. al. [Nature Comm 6, 6313 (2015)] on NbSe₂ strongly back up their claim about impurities and defects resulting in short-ranged CDW order. In addition, the X-ray diffraction paper on TiSe₂ by Kogar et. al. [PRL 118, 027002 (2017)] shows that short-ranged CDWs can survive above the transition temperature in the presence of disorder [Fig. 3 bottom inset]. Lastly, in addition to the papers already cited, the paper by Gleason et. al. [PRB 91, 155124 (2015)] on ZrTe₃ also shows that at around 140K, CDW fluctuations become relevant.

5) The authors first define CDW in the first sentence as “charge density waves” and there is a constant flipping back and forth between the plural and singular noun when using the acronym. A simple solution is to define “charge density wave” as CDW and refer to “charge density waves” as CDWs.

6) The sentence at the bottom of the first page “According to resistivity measurements, the CDW order develops below $T_{\text{CDW}} \sim 63\text{K}$, with wavevector $q_{\text{CDW}} = (0.07, 0, 0.33)$ ” is ambiguous and makes it sound like the resistivity measurements can determine the CDW wavevector.

Reply to Reviewer #1:

(Remarks to the Author):

“The influence of defects on the formation of CDWs is a very interesting topic and still little studied by modern 3rd generation instruments, especially when fluctuations close to the phase transition are taken into account. For this reason, this article deserves all the attention.”

We thank the Reviewer for the time and effort spent assessing our manuscript. We are pleased by the Reviewer’s appreciation of the subject matter that we set out to address. It is indeed our belief that with the development of modern-generation X-ray techniques, prototypical materials deserve to be revisited. Such studies will also make us more confident interpreting related results on other “novel” materials.

The Reviewer raised several technical issues about the content and presentation of our manuscript. We address them below.

“The subject of Friedel Oscillations is particular, little studied by diffraction (a single publication cited in the paper to my knowledge), not easy to observe by diffraction and whose characteristics generate a very original X-ray signature (an asymmetrical contribution in the feet of the $2k_F$ satellite). Evoking the presence of Friedel oscillations to explain the measured asymmetric diffraction pattern requires a much more precise presentation of the foundations of this approach. You must convince the reader that this asymmetrical contribution cannot come from anything else, especially in your case, where the crystal seems to be of poor quality (presence of many rings in the diffraction patterns in Figure 1).”

We thank the Reviewer for raising this important point. Indeed, the first X-ray signature of Friedel oscillations (in a CDW material) was an asymmetrical contribution in the feet of the $2k_F$ satellite reported by Rouzière and coworkers (Ref. [41] in the revised manuscript). We referred to this study in our previous manuscript alongside with the more recent work by Gyenis *et al.*, since we do think they deserve credit. However, we do not think our main observation is merely a “measured asymmetric diffraction pattern” as mentioned by the Reviewer. We clearly demonstrate a coexistence of two peaks that have very different characteristics, including (1) peak-center momentum, (2) momentum width, and (3) intensity, as functions of temperature. None of such comparative information on the two peaks has been reported before, to the best of our knowledge. In addition, we have studied the mesoscopic (slow) dynamics using X-ray photon-correlation spectroscopy and observed a non-monotonic temperature evolution, which is unprecedented. In light of all these experimental information, we think our interpretation based on Friedel oscillations is robust, and grounded in the empirical evidence.

The Reviewer encouraged us to address alternative interpretations, which was indeed

a necessary exercise considering the multiple hypotheses. To do that, we have thoroughly revised the presentation related to results in Figure 2. In the revised text, we refer to the two features as the “broad” and “sharp” peaks, instead of the “FO” and “CDW” peaks as in the previous version, in order to better separate the experimental observations from their interpretations. We now discuss their physical origin in a more elaborating manner by devoting a subsection “**Physical origin of the two signals**” to this purpose. In this subsection, we have added a paragraph that discusses thermal diffuse scattering as a possible alternative origin of the broad peak, and we believe that we have solid evidence to rule this out. The paragraph reads:

A possible alternative interpretation of the broad peak is that it arises from thermal diffuse scattering, since soft phonons (*i.e.*, the Kohn anomaly [38]) are known to prevail at \mathbf{q}_{CDW} near the transition. However, the small momentum transfer of soft X-ray scattering makes it not particularly sensitive to phonons. Even if strong electron-phonon coupling may allow us to see the Kohn- anomalous phonons, the diffuse scattering intensity is expected to reach its maximum at T_{CDW} , where the phonons are softest [38] and the sharp and broad peaks are about to form at the same \mathbf{q}_{CDW} , rather than near T_{LO} , where the two peaks are well-separated. Therefore, while thermal diffuse scattering might be the origin of the weak signals that persist above T_{CDW} [Fig. 2(a)], we rule out the possibility that it produces the broad signatures seen in the range $T_{\text{LO}} < T < T_{\text{CDW}}$.

In regards to the quality of our samples, we would like to discuss the experimental observations (ring-like patterns) highlighted by the reviewer. The “presence of many rings” (Fig. 2) is a consequence of imperfect background subtraction, and it is no indication of poor sample quality. The background has two components. The first one arises from the pixel-dependent electronic noise that is inherent with FCCD operation, which varies slowly and randomly with time. By subtracting a dark image (with the X-ray beam shutter closed) taken right before or after an actual measurement, we can remove most of this noise [Figure R1(a), after the FCCD noise removal]. On the other hand, the FCCD pixel rows had slightly different detection efficiency, and as our diffraction signal was superposed on a non-negligible (but otherwise uniform) fluorescence background, this caused inhomogeneous background on the output images. We minimized such artifacts by subtracting a separately measured fluorescence background, by setting the sample to an off-diffraction angle [Figure R1(b), after the fluorescence background removal]. Unfortunately, neither of these two steps could perfectly remove all the noise or background inhomogeneity, and the remnant effects then became visible as rings in the reconstructed data. As an example, the following figure shows when there is no noticeable signal (data were obtained at 66 K at \mathbf{q}_{CDW} , as in Fig. 2), we still see rings in the reconstructed data [Figure R1(c)].

The momenta carried by our X-rays are too small for producing powder diffraction rings from any possible impurity phases. Moreover, had the rings been caused by diffraction from impurity phases, the center of the rings should be centered at (0, 0,

0), which is clearly inconsistent with the data.

Figure R1. (a) An FCCD image taken in the L -scan (see manuscript) performed at 66 K, after subtracting a dark image taken without X-rays. (b) Result after further subtracting the fluorescence background measured separately (not shown). (c) Data in the (H, L) plane reconstructed from a series of background-subtracted FCCD images, obtained in the same way as in Fig. 2 of manuscript.

In the revised caption of Figure 2, we have added a statement which reads:

Ring-like patterns are due to incomplete FCCD electronic background subtraction, and not related to presence of any impurity phases in our sample.

And in the revised presentation of results in Fig. 2, towards the end of subsection “Coexistence of two distinct signals”, we have added the following sentence and a corresponding new supplementary Figure S5, in order to show that a disordered sample would not allow us to observe the two separate signals.

We emphasize that only high-quality samples allowed us to separate the two signals – measurement on a purposely quenched and disordered crystal yielded only one much broader peak (Fig. S5 in [36]) at all temperatures below T_{CDW} , precluding the identification of the two signals.

Figure S5 (copied from the revised Supplementary Information). Raw FCCD images of the CDW signal of (a) a high-quality crystal at 32 K and (b) a purposely quenched crystal at 27 K. The quench was performed by heating up pristine crystals to 850 °C, followed by cooling in furnace to below 400 °C within one hour.

In addition to the above, we have expanded our discussion related to the observation of two peaks (broad + sharp). We find that the part discussing the experimental data

has significantly improved in scope and detail and we hope that the revisions address the reviewer's remarks in full.

"The data presentation has also to be redone. Figure 1 is incomplete (missing the FWHM, to put in the main text and not in the supplement, it is an essential information) and not very understandable (the variations of the 2 wave vectors are unclear; the addition of a point on the maximum intensity of the diffuse peak is not recommended; is the diffuse peak comes out of the measuring window at low T leading to an artificial decrease of intensity?). There is also missing data concerning the diffuse intensity still present, according to you, above T_c up to 140K."

(We believe that the Reviewer was referring to Figure 2 as being incomplete.)

Following the Reviewer's suggestion, we have expanded Figure 2 by including the most crucial information originally placed in the supplement. The related data commentary in the text has also been revised.

In Figure 2(a), we have removed the red point originally placed on the intensity maxima of the peaks. We confirm that the extraction of the peak intensities is not affected by the broad peak's movement towards the boundary of our data range, and have added the following clarification to the caption of Figure 2(a):

Each image's field of view is sufficiently large (note the larger field of view of the 23 K data set) to include the maxima of both peaks, enabling reliable extraction of their intensities.

Regarding our statement in the previous version that a significant signal may persist up to higher temperatures (about 140 K), we actually stated that it is only an estimation, and is not based on our own data. There are, in fact, related non-resonant X-ray scattering data in the literature that indicate the existence of a signal even up to room temperature. We have revised the corresponding paragraph, which now reads:

(1) A clear signal persists up to at least $64 \text{ K} > T_{CDW}$ [Fig. 2(a)]. At even higher T , the diffraction pattern becomes smeared and weak to the point of being buried under the fluorescence background. Nonetheless, a CDW-related electronic gap [34, 37] and suppression of phonon-damping effects [34, 39] have been observed up to higher temperatures $\approx 140 \text{ K}$, indicative of persisting short-range charge correlations and fluctuations above T_{CDW} . Diffuse scattering signals centered around \mathbf{q}_{CDW} , presumably related to thermally activated soft phonons, have been observed in hard X-ray scattering experiments even up to room temperature [38].

In the meanwhile, we have collected new data in one of our most recent experiments, in order to test the validity of our previous estimation. It turns out that while in one occasion we indeed observed a broad peak at temperatures above T_{CDW} (see Fig. R2a

below), the result seemed to depend on the sample's alignment and/or surface quality over the region illuminated by the X-rays, as the broad peak could not be reproduced on another sample spot which had a considerably stronger long-range CDW peak below T_{CDW} (Fig. R2b). In light of these variations, we concluded that the issue warrants more work and decided to leave out our comment regarding whether we believe (or propose) that a broad resonant X-ray diffraction peak persists to temperatures far above T_{CDW} .

Figure R2. New resonant X-ray diffraction measurements of a ZrTe_3 single crystal. (a) A broad diffuse scattering signal persists to well above T_{CDW} . Data are vertically offset for clarity, and intensities measured at 110 K have been subtracted as background. (b) Similar as in (a), but measured at a different spot on the same crystal. The intensities in (a) and (b) can be compared as they are from a continuous run. The fact that the broad peak is observed when the peak intensity below T_{CDW} (data at 60 K in the figure) is relatively small suggests that the local surface quality and/or impurity density might affect the phenomenon. These measurements were recently done at the REIXS beamline at the Canadian Light Source, hence we have added a sentence in the Acknowledgements to reflect the collaboration behind this new effort.

“In general, the article gives an impression of superficiality because of missing data and a lack of precise explanation of the results (why does this diffuse peak drop in intensity below 50K? Why such temporal fluctuations? Do Bragg reflections show the same asymmetrical contribution - how behave the other satellites? why this broad peak moves in the opposite direction with temperature?), despite the obvious interest of the topic.”

Motivated by the Reviewer's assessment on the data presentation and explanation, we have substantially expanded pertinent parts of our manuscript. To avoid repeating too much of the revised texts here, we kindly refer the Reviewer to pertinent parts of

our revised manuscript, and below we focus on the specific questions raised here.

- Why does this diffuse peak drop in intensity below 50 K?

Please see in subsection “**Physical origin of the two signals**”, the third paragraph, and the fourth paragraph towards the end; and in section “**II. Discussion**”, the third paragraph.

- Why such temporal fluctuations?

Please see in subsection “**Mesoscopic dynamics**”, the last paragraph; and in section “**II. Discussion**”, the first two, as well as the fourth paragraphs.

- Do Bragg reflections show the same asymmetrical contribution - how behave the other satellites?

Due to the limited momenta carried by the soft-X-ray photons, we can only reach one satellite reflection which is the one we present in our manuscript, $\mathbf{q}_{\text{CDW}} = (-0.07\mathbf{a}^*, 0, 0.67\mathbf{c}^*)$. Fundamental Bragg reflections can only be reached with (much less intense) second and higher harmonics of the primary X-ray photons. The peaks are always sharp and they show no asymmetrical profile, see the figure below.

Figure R3. (a and b) Bragg peak (0, 0, 1) FCCD images acquired with 1380 eV X-ray photons at 23

and 66 K, respectively. (c and d) Bragg peak (-2, 0, 3) measured with 2760 eV X-ray photons at 23 and 66 K, respectively. The color scales are linear in intensity in (a-d). The peaks are very sharp – note the substantially smaller pixel range than, *e.g.*, in Fig. S5. (e and f) Theta-2theta scans taken on the (0, 0, 1) and (-2, 0, 3) Bragg reflections, respectively, plotted versus theta and normalized to the maximum of the intensity. Lines are Lorentzian (e) and Gaussian (f) fits to the intensity profiles. The shift between the peak positions at 23 and 66 K is due to (anisotropic) thermal expansion of the crystal lattice.

- Why this broad peak moves in the opposite direction with temperature?

We have removed the word “opposite” in the revised manuscript, as the movements are not exactly opposite between the broad and sharp peaks. Moreover, the movements occur over different temperature ranges for the two peaks. For a more detailed elaboration on the reasons why the broad peak moves *away from* the sharp peak with temperature, please see in subsection “**Physical origin of the two signals**”, the third paragraph; and in section “**II. Discussion**”, the third paragraph.

In summary, we are thankful to the Reviewer for her/his interest in our work, for the thorough inspection of our manuscript, and for the insightful remarks, which greatly helped improve the technical quality and clarity of our manuscript. We hope that the Reviewer will find our revisions and responses satisfactory.

Reply to Reviewer #2

(Remarks to the Author)

“This manuscript applies state-of-art resonant X-ray scattering (REXS) to a charge-density-wave (CDW) material, ZrTe₃, aiming to distinguish between different types of charge ordering. The key result is shown in Fig. 2(a), which describes the temperature dependence of the CDW diffraction peak. Surprisingly, it is found that the main peak is accompanied by a satellite peak at a close-by wave-vector. This interesting experimental fact had not been described or predicted before and deserves publication.

At the same time, I think that the manuscript contains many unjustified claims and conjectures and requires a complete rewriting”

We thank the Reviewer for the time and effort spent examining our manuscript. We are pleased that the Reviewer appreciated one of our key findings presented in Fig. 2. In the meantime, we feel that part of the Reviewer’s critique might arise from our failure to properly convey the scientific message and the representation of the related literature. We have taken significant steps to rewrite and expand a substantial part of our manuscript, aiming to improve the clarity of our presentation. In the following, we address the Reviewer’s comments in detail.

“1. The paper is framed in the context of Cuprate high-temperature superconductor, where REXS have become very popular thanks to the detection of a short-ranged CDW order by Ghiringhelli et al (Science 2012). Since then, hundreds of papers (including tens of Science and Nature papers) were written trying to establish whether the signal is due to a true competing CDW order, or merely, to Friedel oscillations (FO). The present manuscript claims that the double peak observed in ZrTe₃ helps clarifying this debate: They associate the main and satellite peaks, respectively, to CDW and to FO. This situation is apparently similar to the high-magnetic-field measurements of Gerber et al (Ref. [7]), where two distinct diffraction peaks were found. However, there are two main differences:

- a. In Cuprates, the short-ranged FO and the long-ranged CDW appear at different temperatures and/or magnetic fields, while in ZrTe₃ they appear exactly at the same temperature.
- b. In Cuprates, FO and CDW appear at well-separated wave-vectors (in the “c” axis component), while in ZrTe₃ they appear one next to the other.

Because of these two key differences, I do not think that the present study is directly relevant to Cuprates, nor that the two peaks are associated with FO and CDW oscillations.”

We do consider the currently intense interest on CDW phenomena in the cuprates an important motivation for our study, and we believe that we have obtained new and sound insights from a model CDW system, which will help advance our knowledge on

CDWs in general, and on the role of disorder in particular. We had, however, no intention to imply that there is a one-to-one correspondence between our results and those in the cuprates. The Reviewer pointed out that there are major differences between ZrTe_3 and cuprates in high fields (magnetic fields or strains) concerning the presence of two distinct diffraction peaks, which we fully agree with. In fact, we did not even attempt to comment on the common presence of two distinct diffraction peaks in both ZrTe_3 and YBCO (in high fields), and would like to thank the Reviewer for drawing our attention to this possibility, which we find intriguing and may want to explore in the future.

To provide a clearer context for the readers, along the lines of what discussed above, we have revised the closing paragraph, which now reads:

In light of our results, it is interesting to notice that when long-range CDWs form in the cuprate superconductors [7-9], the sharp diffraction peak is accompanied by a much broader peak associated with short-range CDWs. While this coexistence of two signals bears some similarity to our finding in ZrTe_3 below T_{CDW} , the two systems are very different both in the signals' amount of separation in \mathbf{q} , and in their dependence on temperature (and external fields). Hence we do not think a direct analogy between the CDWs (and the FOs) in ZrTe_3 and the cuprates should be drawn at this point. Yet empirically, it does seem that even the rather long-range CDWs studied recently in $\text{La}_{1.875}\text{Ba}_{0.125}\text{CuO}_4$ lack the phenomena that we have seen: only one peak (considerably broader than ours) and one characteristic temperature were observed, and the speckle patterns were found to be always static at low temperatures [33] and with memory effects up to high temperatures [47]. The counterpart of our finding, namely, the role of disorder in charge-order formation in the strong-coupling limit, to which the cuprates belong, deserves further experimental scrutiny.

Regarding the closing of the Reviewer's remark, we agree that our results do not lead to the conclusion that the two peaks in the cuprates are associated with FOs and CDWs (perhaps that is what the Reviewer meant to say?), however, we do not think that the lack of direct correspondence with the cuprates would invalidate our interpretation of the two signals in ZrTe_3 as due to FOs and CDWs.

"2. The paper makes contradictory claims about the CDW transition temperature. These claims are based on linear extrapolations of the REXS signal and suggest transition temperatures ranging from 52K to 59K. This large uncertainty needs to be contrasted with the sharpness of the resistivity measurement, which shows a pronounced lambda peak at 63K. Because the latter experiment gives a much sharper signal, I do not think that the linear extrapolation performed by the authors has any scientific relevance. (Just for fun, I attach a plot in which a different linear extrapolation gives a transition temperature of 64K). Similarly, I do not understand why the critical temperature should correspond to a saturation of the FWHM width. This saturation implies that the CDW signal is full developed: It always occurs much below the critical

temperature (where the long-range component of the CDW starts developing and is still accompanied by strong short-range fluctuations). Finally, the conjecture that the diffraction signal may be present already at 140K is not supported by the experimental findings and should be erased from the text.”

We thank the Reviewer for raising these very important points. In the original manuscript, there might have indeed been some confusion regarding the exact transition temperatures that we intended to claim or compare. In our revision, we have clarified how we define and refer to the characteristic temperatures, by explicitly introducing T_{CDW} (= 63 K) and T_{LO} (= 56 ± 3 K). We believe that our revised manuscript presents a self-consistent and convincing analysis. We try to address the Reviewer’s previous remarks in greater detail here below.

We do believe that there are two, rather than one, characteristic temperatures seen in our measurements. The resistivity measurement (especially in the T derivatives) does give a well-defined temperature of 63 K, which has been regarded as the CDW transition temperature in previous works, so we call it T_{CDW} in order to keep the same terminology as in the literature. This temperature is not only seen in resistivity – previous hard X-ray diffraction saw it as the canonical CDW transition temperature, below which there is a rapid increase of the satellite intensity (the work is cited as Ref. [38] in our revision); we also see a rapid increase of the broad peak’s intensity below this temperature, and it is the temperature below which the two peaks in Figure 2 start to separate.

However, the sharp peak tells a different story by showing that there exists a somewhat lower characteristic temperature, around 56 K, which we call T_{LO} (LO: long-range order) in our revision. This temperature has not been reported before, and we believe that it is because our method of resonant X-ray scattering is exclusively and highly sensitive to the conduction electrons’ spatial organization, which is unique among other techniques. For instance, hard X-ray diffraction is mainly sensitive to lattice deformations, and transport measurements are sensitive (in the Drude model) to both the average number of free electrons and their scattering rate, the latter of which can also be strongly affected by lattice deformations. Hence, it is not a total surprise that we are able to see the new T_{LO} , in addition to seeing T_{CDW} as reported elsewhere.

Back to the experimental data: The existence of T_{LO} distinct from T_{CDW} is established by a chain of evidence, not simply by a linear extrapolation. Nonetheless, regardless of the quantitative approach to the sharp peak’s intensity data (*e.g.*, in Fig. 2(d)), the observed behavior (*vs. T*) is qualitatively very different from that of the broad peak, which increases rapidly right below T_{CDW} . The sharp peak’s intensity variation with T clearly speaks for a transition temperature below 60 K, as the main increase occurs only below a temperature (our T_{LO}) that is lower than 60 K, and the little tail that extends to above 60 K can be understood as either due to a small transition-

temperature inhomogeneity, or because of cross-over behavior in the presence of an “external” field conjugate to the order parameter in a Landau-theory language. Given the existence of the rapidly-increasing broad peak below T_{CDW} , which is very close by in \mathbf{q} to the sharp peak and can hence indicate the presence of such a conjugate field, we believe that the second explanation, *i.e.*, a cross-over behavior, is most natural here. We would not make conclusions by zooming into the tiny tail, and we think that it is incorrect to consider that “the CDW signal is fully developed” (at T_{LO}) – over 95% of the total intensity of the sharp peak is still missing at T_{LO} .

We agree with the Reviewer that in a canonical 2nd-order phase transition, the correlation length, as well as the order parameter (intensity in our case), are expected to exhibit a major increase below the transition temperature. It can be argued that T_{LO} cannot be pinpointed with high accuracy (hence the relatively large uncertainty we have assigned to its estimated value, 56 ± 3 K) in light of the expected concurrent increase in the two quantities. However, we can safely conclude that it cannot be as high as T_{CDW} , which we can clearly identify the same data set from the broad peak (thus there is no issue here about inaccurate temperature reading).

Altogether, our whole data set consistently shows that T_{LO} is distinct from T_{CDW} and that it has a handful of empirical manifestations and consequences, in addition to the intensity increase of the sharp peak: (1) the sharp peak’s widths drop rapidly near, and saturate below, this temperature; (2) the sharp peak’s \mathbf{q} center becomes no longer sensitive to T below this temperature; (3) the broad peak’s intensity decreases approximately below this temperature; and (4) the speckle patterns are static above but dynamic below this temperature.

In summary, we agree with the Reviewer that our previous presentation was not particularly coherent in reporting transition temperatures, yet we firmly believe in the existence of distinct T_{LO} and T_{CDW} , which are now clearly defined and discussed in our revised manuscript, along the lines of arguments outlined above. We thank the Reviewer for motivating us to clarify this important and new observation.

The Reviewer remarked that our conjecture that diffraction signal may be present already at 140 K is not substantiated by our own data, which we agree. Recently, we have collected new data in order to test the validity of our previous conjecture. It turns out that while in one occasion we indeed observed a broad peak at temperatures above T_{CDW} (see Fig. R4a below), the result seemed to depend on the sample’s alignment and/or surface quality over the region illuminated by the X-rays, as the broad peak could not be reproduced on another sample spot which had a considerably stronger long-range CDW peak below T_{CDW} (Fig. R4b). In light of these variations, we concluded that the issue warrants more work and decided to leave out our comment regarding whether we believe (or propose) that a broad resonant X-ray diffraction peak persists to temperatures far above T_{CDW} .

Figure R4 (same as Fig. R2). New resonant X-ray diffraction measurements of a ZrTe_3 single crystal. (a) A broad diffuse scattering signal persists to well above T_{CDW} . Data are vertically offset for clarity, and intensities measured at 110 K have been subtracted as background. (b) Similar as in (a), but measured at a different spot on the same crystal. The intensities in (a) and (b) can be compared as they are from a continuous run. The fact that the broad peak is observed when the peak intensity below T_{CDW} (data at 60 K in the figure) is relatively small suggests that the local surface quality and/or impurity density might affect the phenomenon. These measurements were recently done at the REIXS beamline at the Canadian Light Source, hence we have added a sentence in the Acknowledgements to reflect the collaboration behind this new effort.

Yet still, we consider it potentially misleading to leave the readers with the impression that charge correlations are completely gone at temperatures immediately above T_{CDW} , as precursors of the CDW state have been observed in previous experiments, in some cases even up to room temperature. We have revised the pertinent statements to be more specific:

Nonetheless, a CDW-related electronic gap [34, 37] and suppression of phonon-damping effects [34, 39] have been observed up to higher temperatures ≈ 140 K, indicative of persisting short-range charge correlations and fluctuations above T_{CDW} . Diffuse scattering signals centered around \mathbf{q}_{CDW} , presumably related to thermally activated soft phonons, have been observed in hard X-ray scattering experiments even up to room temperature [38].

“3. As mentioned above, the experiment shows that the two peaks are born at the transition temperature of the CDW phase, $T=64 \pm 1$ K. This finding is not consistent with the schematic plot of Figure 4, where the FO start developing at high temperatures and progressively become longer and longer ranged. Furthermore, the picture presented in Figure 4 does not explain the emergence of a satellite peak in the

X-ray scattering at low temperature (which is the main experimental finding of this paper).”

This remark links back to the previous point. We apologize for the confusion. We have now clarified these aspects in our revised manuscript. Now panel (a) of Figure 4 corresponds to $T_{LO} < T < T_{CDW}$ (62 K in Fig. 3), (b) to T comparable to or slightly below T_{LO} (47 K and 56 K in Fig. 3) and (c) to T much lower than T_{LO} . In addition, in (b) and (c) we have added an exaggerated 10% difference between the periodicity of the red (FOs) and grey (CDWs) modulations, in order to reflect our experimental observation of the separated sharp and broad (“satellite” according to the Reviewer) peaks in Fig. 2. The difference is highlighted by a blue circle in panel (c) of the revised Figure 4.

“4. The authors claim that Ref. [38] was not able to separate CDW and FO oscillations. I do not agree with this statement. See in particular Fig. 1 there, where two peaks at wavevectors q_c and q_s are detected. I think that the authors should give more highlight to this earlier publication, which demonstrated for the first time that X-ray scattering can probe Friedel oscillations!”

This is another passage where we misrepresented our interpretation of prior studies – here we meant to remark that the previous authors did not observe (1) the two signals as fully separated peaks and, more importantly, (2) their process of separation versus temperature. We agree with the Reviewer that Ref. [38] (now Ref. [41]) deserves to be highlighted as the first X-ray scattering observation of Friedel oscillations. Along with the highlight, we feel that we ought to also point out an important difference between the previous results and ours. Our revised discussion of this previous work now reads:

In light of our results, the previously observed asymmetric profile of the CDW satellite reflections, similar to our data at 62 K (Fig. S2 in [36]), may indeed originate from sum of two (CDWs and FOs) signals [41], but the high impurity concentration of the previous sample did not allow the authors to observe a clear and progressive separation of the signals, nor to address their relation on the verge of CDW formation. Being able to observe such progressive separation but without noticeable change in the width of the broad peak with cooling, we further demonstrate that the separation we observe (Fig. 2) is unrelated to the (short) correlation length of the FOs, which was used to explain the previously observed separation [41].

“5. One possibility is that the satellite peak is due to FO induced by the domain walls of the CDW order (This may explain the non-monotonous temperature dependence). However, how can one explain the tiny momentum difference between the main CDW peak and the satellite peak? I find it hard to believe that this is due to the nesting between two parallel bands, along the lines of Ref. [19-21].”

In our data, the broad peak (referred to by the Reviewer as the “satellite” peak) rapidly increases below 63 K. If this peak arises from FOs induced by the domain walls of the

CDW order, the initially formed domain walls should be frequently activated as the temperature is high, but that would be inconsistent with our observation of static speckles at 62 K. For precisely this reason, we believe that regular domain walls of the CDWs only form at a somewhat lower temperature, around and below T_{L0} as mentioned in our response to one of the Reviewer's earlier comments.

In our revised manuscript, we have improved our discussion related to using Fermi-surface nesting to explain the momentum difference between the main CDW peak and the broad (satellite) peak. The related schematics in Figure 1(d) have also been improved, where one can see the nesting vectors' difference as indicated by the two colored arrows. We hope that the Reviewer will find our revised discussion of this aspect more complete and convincing.

“To summarize, I think that this scientific work is very timely and constitutes an important advancement of the field. REXS has recently become one of the leading probes to describe strongly correlated materials, and high-temperature superconductors in particular. Hence, it is very important to apply REXS to other known materials, to understand the limitations and advantages of this probe. This paper offers a much-needed independent test of this technique. However, the text of the manuscript has to undergo a major revision before being considered in Nature Communications.”

We very much appreciate the Reviewer's appraisal of our work's timeliness and relevance to on-going research efforts. We have made major revisions to our manuscript, many of which were motivated by the Reviewer's constructive criticisms and suggestions. They are indicated in blue fonts in the auxiliary .pdf file.

Reply to Reviewer #3

(Remarks to the Author)

“The paper by Yue et. al. represents an interesting study on the charge density wave (CDW) compound ZrTe₃, showing that there are two contributions to the CDW superlattice reflection. One comes from the long-range order, which shows up below ~55K, while the other weaker and broader reflection, shows up at a different wavevector and at slightly higher temperature (~63K). The authors interpret this secondary reflection as arising from “Friedel oscillations” seeded by impurities. The authors present supplementary evidence of their interpretation using intensity correlation measurements, showing time dependent signals only in the intermediate temperature range below the transition temperature. Overall, the paper is written well, and most points are made with clarity. If I understand the text correctly (which I am not sure that I do, see the comments below), the authors are likely correct in their interpretation of their data. Overall, the work is solid, the technical achievements in this work are noteworthy (the momentum resolution achieved and the coherent diffraction speckle), and the observations are interesting enough to warrant publication in Nature Communications if some major revisions are made.”

We thank the Reviewer for the time and effort spent assessing our manuscript. We are pleased that the Reviewer appreciated our work and considered it interesting, solid, and technically noteworthy. We are grateful to the Reviewer for the thorough inspection and for the constructive suggestions on how to improve the manuscript. We address the Reviewer’s remarks in the following.

“1) My main point of confusion, which was alluded to above, concerns what the authors refer to as ‘Friedel oscillations’. Naively, I imagine ‘Friedel oscillations’ as something unassociated with a charge density wave but can arise in any Fermi gas/liquid. However, both Friedel oscillations and a charge density wave instability occur nominally at twice the Fermi wavevector. What I do not understand from the text is whether the authors are trying to draw a distinction between ‘Friedel oscillations’ and an impurity-pinned short-ranged CDW or whether they are referring to them as the same phenomenon. My comments for either interpretation are presented below.

a. If the authors are trying to distinguish the two as different physical phenomena, a couple sentences need to be written in the introduction explaining how they are different. Furthermore, then there would need to be strong evidence in the data that is able to differentiate between the two, which would need to be discussed in the text after the data is presented. Naively, I do not necessarily see the difference between the two in the data presented by the authors. Does resonant soft X-ray scattering help distinguish the two somehow in a way that I am unaware?

b. If the authors are referring to Friedel oscillations and the short-ranged CDW as the same physical phenomenon, they have chosen a rather unconventional terminology, which is likely to confuse other readers as well. In my reading, this is what I understood

the paper to mean, and I think this interpretation of the data is likely to be correct. To those used to studying conventional charge density waves, one usually refers to the phenomenon as a short-ranged CDW or a localized impurity-pinned or defect-pinned CDW. Indeed, if the authors are equating the two physical phenomena, I would suggest that the authors change their terminology. Indeed, the STM paper by Arguello et. al. [PRB 89, 235115 (2014)] on NbSe₂ discusses at length why what they observe should be interpreted as a short-ranged CDW stabilized by defects rather than being equated with Friedel oscillations.”

The point raised by the Reviewer is a very important one, even though the issue at stake here might boil down to a matter of terminology. By thinking closely about the Reviewer’s questions during our revision, we believe that we have reached a more coherent and better-defined understanding of our results altogether.

To summarize some of the key revisions that we have made, we do agree with the Reviewer that, in many circumstances, it is intrinsically difficult, if not impossible at all, to differentiate Friedel oscillations and impurity-pinned short-ranged CDWs. This is particularly true from the symmetry point of view, as the two phenomena break the original lattice translational symmetry in precisely the same manner (same $\mathbf{q} = 2*\mathbf{k}_F$). They even share the same lack of mesoscopic dynamics (when the impurities do not move).

But in our case, thanks to the slight warping of the nested Fermi surfaces in momentum directions perpendicular to the nesting momentum, which can very well be present in other CDW systems, there is a gradual departure between the wave vectors of the CDWs and the FOs below the temperature commonly recognized as the CDW transition temperature ($T_{CDW} = 63$ K in our case). In the temperature range between $T_{LO} = 56 \pm 3$ K and T_{CDW} , which we define with improved clarity in the revised manuscript, the pinned CDWs and the FOs are characterized by slightly different wave vectors as shown in Fig. 2. In this way, they can actually be distinguished – the broad peak corresponds to FOs, and the sharp peak to pinned CDWs. According to our interpretation, both peaks grow below T_{CDW} because (1) the FOs receive a positive feedback from the lattice, and (2) it is the lattice-deformation-enhanced FOs that help pin the CDWs most effectively, not the bare impurities themselves. Reason (2) also explains why the FOs grow more rapidly with cooling below T_{CDW} than the pinned CDWs: the pinned CDWs observed between T_{CDW} and T_{LO} , although sufficiently strong to dominate over the FO signal in terms of producing a diffraction pattern with coherent (static) speckles, merely represent a cross-over behavior associated with a transition at the lower temperature of T_{LO} .

Another way of looking at the results in Fig. 2 is the following: the pinned-CDW signal must show continuation (as a function of T) to the long-range CDW signal at, *e.g.*, 23 K, which is unquestionably the sharp peak. Therefore, one needs a different explanation for the broad peak (at any temperatures). We think it is most natural to

attribute it to FOs, as otherwise one needs two versions of pinned CDWs between T_{LO} and T_{CDW} (note that the sharp peak still dominates the total diffraction signal, so the speckles associated with the sharp peak must be quite static in order to explain our observation at 62 K in Fig. 3, *i.e.*, it is “pinned”), which is a bit of an artificial scenario. Similarly, we can rule out thermal diffuse scattering as a possible alternative explanation for the broad peak (also per the request of another reviewer; see our response to Referee 1’s first criticism, and the paragraph on page 4 of our revised manuscript beginning with “A possible alternative interpretation ...”).

Our interpretation is also compatible with discussions in previous literature, *e.g.*, in Arguello *et al.*, PRB 89, 235115 (2014), by improving our understanding of the CDW-pinning mechanism. In our picture, the pinning is assisted by a frozen lattice soft mode (leading to a permanently deformed lattice), hence it is distinct from the “STM version” of Friedel oscillations (also known as QPI, which has energy-momentum dispersion). We have added the above reference (along with two other related papers suggested by the Reviewer) to our revised discussion.

The most relevant revisions in response to the above remarks by Reviewer are from the second to the fourth paragraphs of the **Discussion** section, although some of the ideas have been mentioned elsewhere in the revised manuscript. We hope that the Reviewer will find our revisions satisfactory.

“Other than this rather major issue in the paper, I have some other minor comments which the authors should address:

1) Nowhere in the main text do the authors talk about the length scales associated with either CDW. Since the authors consistently refer to the peaks as “broad” and “sharp”, it would be beneficial to the reader to understand the length scales involved, even if just a lower bound on the length scale of the sharper peak is presented (which is likely limited by resolution). The data is ultimately presented in the supplement, but it would be helpful to have an estimate in the main text.”

We have expanded Figure 2 by incorporating some of the supplementary figures into it. The saturated narrow widths of the sharp peak (below T_{LO}) are not resolution-limited in all directions. We have added the following descriptions of the data in the revised text:

(3) We identify a new characteristic temperature, $T_{LO} = 56 \pm 3$ K. Below T_{LO} , the sharp peak's \mathbf{q} position becomes nearly insensitive to T [Fig. 2(e-f)], and its \mathbf{q} widths become narrow and saturated [Fig. 2(g-h) and Fig. S3 in Ref. 36], amounting to minimal correlation lengths of about 9000, 1000, and 400 Å along \mathbf{a}^* , \mathbf{b}^* , and \mathbf{c}^* , respectively. These indicate that a truly long-range CDW order forms and the associated charge-modulation periodicity reaches a stationary stage below T_{LO} . In contrast, the broad peak continues to evolve in its \mathbf{q} position, and remains broad (broader than the sharp peak by a factor of 3-6) below T_{LO} .

“2) The authors may want to draw the nesting vector on the Fermi surface in Fig. 1(c). To those not familiar with the material, this would be beneficial.”

We have revised Fig. 1 and added arrows to indicate nesting vectors in Fig. 1(d). They also allow us to better discuss the separation between the two peaks in Fig. 2. We thank the Reviewer for this suggestion.

“3) Some plots refer to a peak amplitude and some refer to a peak intensity. I would use the latter terminology, as it is less ambiguous. Unless, of course, the authors do mean to draw a distinction, in which case I would suggest clarifying what is meant.”

We have changed all instances of “amplitude” to “intensity”, following the Reviewer’s suggestion.

“4) The authors do not cite several references which appear to be highly relevant to their observations. The STM papers by Arguello et. al. [PRB 89, 235115 (2014)] and Chatterjee et. al. [Nature Comm 6, 6313 (2015)] on NbSe₂ strongly back up their claim about impurities and defects resulting in short-ranged CDW order. In addition, the X-ray diffraction paper on TiSe₂ by Kogar et. al. [PRL 118, 027002 (2017)] shows that short-ranged CDWs can survive above the transition temperature in the presence of disorder [Fig. 3 bottom inset]. Lastly, in addition to the papers already cited, the paper by Gleason et. al. [PRB 91, 155124 (2015)] on ZrTe₃ also shows that at around 140K, CDW fluctuations become relevant.”

We thank the Reviewer for drawing our attention to all these highly relevant previous works. All of them have been added to the references and briefly discussed in our revised manuscript.

“5) The authors first define CDW in the first sentence as “charge density waves” and there is a constant flipping back and forth between the plural and singular noun when using the acronym. A simple solution is to define “charge density wave” as CDW and refer to “charge density waves” as CDWs.

6) The sentence at the bottom of the first page “According to resistivity measurements, the CDW order develops below $T_{CDW} \sim 63K$, with wavevector $q_{CDW} = (0.07, 0, 0.33)$ ” is ambiguous and makes it sound like the resistivity measurements can determine the CDW wavevector.”

We have made these corrections suggested by the Reviewer. Thank you!

Reviewers' comments:

Reviewer #1 (Remarks to the Author):

The new version of the paper is much better and I would like to acknowledge the effort of the authors to make the article more intelligible, with better presentation of measurements, much more convincing. I remain convinced that strongly pinned areas, more disordered, are at the origin of the presence of this second peak. However, I remain sceptical about an interpretation in terms of FO. This pure screening effect should, in theory, disturb only the phase of the wave and not its wavelength. However, one cannot rule out a different band structure in these areas (by the way, you should widen the area around the blue circle in Figure 4C). It seems to me that one cannot exclude either the presence of topological defects in these regions, like solitons, as invoked in CDW materials [for example, Phys. Rev. B 94, 201120]: a phase jump of 2π also causes a shift of the norm of the $2k_F$ wave vector, which does not disagree with your measurements either. In conclusion, I will rather favor a publication with some restrictions on the interpretation which seems to me too much focussed on FO. A broader discussion mentioning several scenarios is necessary.

Reviewer #2 (Remarks to the Author):

The authors have significantly improved the manuscript and addressed my critiques, with the exception of one fundamental point, concerning validity of their interpretation.

The experiment by Yue et al reveals two peaks in the REXS scattering of $ZrTe_3$, with a distinct temperature and momentum dependences. This observation is interesting and deserves publication. However, I think that the interpretation of the two peaks as CDW and Friedel oscillations, respectively, is not imperative: The data is open to many other possible interpretations as well. For example, it is interesting to observe that the broader peak appears at around 63K, where resistivity measurement show a CDW phase transition. The temperature dependence of this peak fits well the expected behavior of the order parameter at a symmetry-breaking phase transition. In contrast, the sharp peak appears more gradually and seems a secondary effect, such as a commensurate lattice distortion induced by the electronic CDW. As mentioned by the Referees, other plausible explanations exist as well.

Hence, although I support the publication of this article in Nature Communications, I recommend that the editor makes sure that the distinction between the experimental result and the offered interpretation is made clear in the title, abstract, introduction, and conclusion of the article. Many readers focus on these parts only and it is important not to deceive them by letting them think that the article proves without any doubt that FO and CDW coexist in $ZrTe_3$. For example, "Distinct fingerprints" could be changed to "Possible indications", "These observations demonstrate" to "We interpret these observations", and so on so forth.

Reviewer #3 (Remarks to the Author):

I still like the paper – the data is noteworthy on two fronts. The two peaks in the diffraction plots and the time-dependent speckles are both observations that are worthy of publication in Nature Communications. However, I still have some reservations about the paper in its current form and with some aspects of the interpretation.

Before I discuss that, however, I should mention that parts of the manuscript are greatly improved. The authors have included length scales in the paper, which has greatly improved its accessibility and the general understanding of the experimental phenomenology. Furthermore, the discussion of the two peaks has been improved with the “broad” and “sharp” peak terminology.

Moving onto the interpretation of the experimental results – I believe that there are still substantial problems that need to be resolved in this regard. It is possible that I am partially at fault for this, as upon re-reading my previous review, it seems like what I said may not have been completely clear to the authors. If this was the case, I do apologize to the authors.

In my previous review, I was not suggesting that the FOs and short-ranged CDWs pinned by impurities were the same phenomenon or difficult to tell apart necessarily. I believed that the authors were conflating the two physically different concepts.

The question I wished to raise in the previous review ultimately boils down to the following: Is the observation of the “broad” peak clearly the result of FOs? With the data presented, I cannot convince myself that this is the case. It seems to me more likely that the observations seen here should be attributed to short-ranged CDWs pinned by impurities. If the authors do not think that this is the case, I think it is necessary that they clarify why, which I don’t think they have done adequately in the current manuscript or in the response to my previous review.

Additionally, there are a couple observations which go against their interpretation based on FOs.

First, as raised by one of the other reviewers in the last round, the fact that the peaks show up at the same temperature seems contrary to what one would expect of FOs. In Ref. 41, Pouget et. al. claim to see a temperature-independent signal due to FOs.

Second, as the authors themselves state in the manuscript, the FOs should result in the observation of a contour and not a sharp peak in reciprocal space. I don’t fully understand their explanation in the text as to why this wouldn’t be the case.

Ultimately, what I am trying to say is that one can probably attribute most of the observations here to CDW fluctuations that get pinned by impurities. For instance, it wouldn’t be surprising for these CDW fluctuations to get pinned by impurities close to, but above TLO. Even below TLO, near the impurity sites, the CDW texture can still get warped leading to a slightly different wavevector in the vicinity of the impurities. As the CDW becomes more rigid with decreasing temperature, the “broad” peak would thus weaken. This would also explain why the peak persists below TLO.

I have a two minor comments which the authors might want to take into consideration:

1) The paragraph on pg. 4 that begins “We believe that the discrepancy is caused by...” claims that the lattice somehow undergoes long-range ordering at a higher temperature than the electrons (long-range order is clearly observed at 63K in the paper by Hoesch). This claim is highly dubious – it is electron-phonon coupling that is driving the transition in the first place.

2) The paragraph on pg.4 that begins “A possible alternative explanation...” is not necessary. It seems obvious from the temperature and wavevector dependence that the peak does not arise due to thermal diffuse scattering.

Ultimately, the previous revision did improve the paper in some parts but worsened it in others. The data in the paper is great and deserves publication, but the authors need another revision to sort out

a clear and consistent interpretation of their results. From what I gather, the authors somehow see the FOs as related to the CDW formation. I regard FOs and CDWs pinned by impurities as two distinct physical phenomena. If the authors are convinced that what they are seeing come from FOs, they must explain why with more clarity and more convincingly.

Response to Reviewer #1:

(Remarks to the Author):

“The new version of the paper is much better and I would like to acknowledge the effort of the authors to make the article more intelligible, with better presentation of measurements, much more convincing.”

We thank the Reviewer for the time and effort spent assessing our work. It is gratifying for us to see that our previous revision has been appreciated by the Reviewer.

“I remain convinced that strongly pinned areas, more disordered, are at the origin of the presence of this second peak.”

We fully agree with the Reviewer that the broad peak must come from regions in our sample that are relatively more disordered. This understanding is in part the reason why we believe that the broad peak arises from Friedel oscillations (FOs). While the FO part of our interpretation remains to be criticized by the Reviewer (and by the other two reviewers) for lacking a thorough experimental proof, the above phenomenological understanding has been agreed upon in common. Hence in our revised manuscript, we further emphasize this part of our interpretation, and explicitly point out the advantages and uncertainties associated with our continuation to attribute the broad peak to FOs.

For example, we have modified the third paragraph under the “Physical origin of the two signals” sub-section, which is the first place in our manuscript where the broad peak is attributed to FOs. The beginning of the paragraph now reads:

As discussed towards the end of the previous sub-section and supported by further evidence in the next sub-section, the broad peaks signal ought to come from regions in our sample that contain disorder. With alternative possibilities to be discussed later, here we consider it most likely that the signal arises from standing waves created by self-interfering electrons scattered off a localized potential, *i.e.*, FOs, or is closely related to FOs.

And we write at the end of the same sub-section:

To end this sub-section, we emphasize that our direct observation of coexisting sharp and broad diffraction signals, along with their distinct temperature evolution, is new and revealing. They are the signatures of pristine and disorder-induced charge correlations, respectively. Hereafter, we present another method to highlight the influence of disorder by investigating the mesoscopic dynamics, which must be different between spontaneous and disorder-induced effects.

“However, I remain skeptical about an interpretation in terms of FO. This pure screening effect should, in theory, disturb only the phase of the wave and not its wavelength. However, one cannot rule out a different band structure in these areas (by the way, you should widen the area around the blue circle in Figure 4C). It seems to me that one cannot exclude either the presence of topological defects in these regions, like solitons, as invoked in CDW materials [for example, Phys. Rev. B 94, 201120]: a phase jump of 2π also causes a shift of the norm of the $2kF$ wave vector, which does not disagree with your measurements either. ”

We thank the Reviewer for bringing up this important point. We agree with the Reviewer that there exists, in principle, alternative interpretations for the broad peak. The options offered by the Reviewer are thoughtful, and we feel that they indeed deserve to be mentioned. With respect, we wish to keep our original interpretation based on FOs as the “most likely” one, because it readily explains all our experimental observations. We have done some literature search and found that the alternatives suggested by the Reviewer need to be supplemented by additional quantitative details, which are not obvious *per se*, in order to be consistent with our experimental observations. In our revision, the following paragraph has been added to the Discussion section:

There are alternative, albeit in our opinion more remote, ways to understand the origin of the broad peak and its momentum detuning from the sharp peak. One possibility is that it arises from *directly* induced CDWs in the immediate neighborhood of disorder sites. In contrast, in both this scenario and our original interpretation based on FOs, the sharp peak at $T_{LO} < T < T_{CDW}$ arises from CDWs that are *indirectly* induced and pinned at distance from the disorder, since it dominates the signal and produces the static speckles, and because it continuously evolves below T_{LO} into the peak of truly long-range CDWs. Regarding the detuning in \mathbf{q} between the two peaks, although the presence of impurity in $ZrTe_3$ may locally modify the electronic structure [40] or create topological defects in the CDWs, it remains to be investigated why the detuning effect gradually develops below T_{CDW} and is not present at T_{CDW} .

This paragraph is followed by another paragraph that discusses thermal diffuse scattering, which we rule out as a possible origin of the broad peak. (In the previous version of our manuscript the discussion about thermal diffuse scattering was introduced at an earlier place.)

Regarding the specific reference [PRB 94, 201120] mentioned by the Reviewer, we have studied the paper and found it not to be particularly relevant to our case, because those “solitons” appear in the sliding state of CDWs driven by an applied electric field. We have thus refrained from mentioning this work as an example for topological defects. Other types of topological defects are of course still possible (such as in *Science* 333, 426; but the context there is also very different from ours so we have decided not to mention it in our manuscript).

Following the Reviewer’s suggestion, we have revised Figure 4c by enlarging the blue circle and adding arrows to point at the (slightly) different periodicity of the two types of charge modulations.

“In conclusion, I will rather favor a publication with some restrictions on the interpretation which seems to me too much focused on FO. A broader discussion mentioning several scenarii is necessary.”

We thank the Reviewer again for the favorable assessment of our work and the constructive suggestions. We trust that the Reviewer will find our interpretation and discussions to be more balanced in our revision.

Response to Reviewer #2

(Remarks to the Author):

“The authors have significantly improved the manuscript and addressed my critiques, with the exception of one fundamental point, concerning validity of their interpretation.”

We thank the Reviewer for the time and effort spent assessing our work. It is gratifying for us to see that our previous revision has been appreciated by the Reviewer. We address the Reviewer’s remaining reservation in the following.

“The experiment by Yue et al reveals two peaks in the REXS scattering of ZrTe₃, with a distinct temperature and momentum dependences. This observation is interesting and deserves publication. However, I think that the interpretation of the two peaks as CDW and Friedel oscillations, respectively, is not imperative: The data is open to many other possible interpretations as well.”

We appreciate the Reviewer’s interest in our experimental observation. The point raised by the Reviewer here is valid and, in fact, shared by the two other reviewers. Collectively, the reviewers’ feedbacks have motivated us to modify our manuscript by consistently tuning down the emphasis on the FOs and, instead, be more explicit about the general influence of disorder.

Among other changes which we will mention later in response to the Reviewer, we are now devoting two paragraphs in the Discussion section, beginning with “There are alternative ...” to present possible alternative interpretations for the broad peak, followed by a revised paragraph which discusses how the physics behind the broad peak leads to the development of the sharp peak.

“For example, it is interesting to observe that the broader peak appears at around 63K, where resistivity measurement show a CDW phase transition. The temperature dependence of this peak fits well the expected behavior of the order parameter at a symmetry-breaking phase transition. In contrast, the sharp peak appears more gradually and seems a secondary effect, such as a commensurate lattice distortion induced by the electronic CDW.”

We appreciate the Reviewer’s close attention to the experimental data and, in a way, we understand why the Reviewer tends to refer to the sharp peak as a “secondary effect” – it develops later (upon cooling) than the broad peak. However, taking all the experimental results

together, we have to disagree with the Reviewer's opinion that the sharp peak is secondary or due to a commensurate lattice distortion (which is not seen by hard X-rays, see Ref. [38]) induced by the electronic CDWs, the main reason being that it is actually the sharp peak that carries over 99% of the total diffraction intensity at 30 K (see Fig. 2d in our manuscript). As resonant X-ray diffraction is most sensitive to electron density modulations in the conduction band, the giant intensity carried by the sharp peak must mean that it represents the primary order parameter of the CDWs.

In line with the Reviewer's observation that the sharp peak initially has a gradual intensity increase below 63 K, we have attributed this behavior to a crossover of the CDWs under the influence of the charge modulations (FOs in our "most likely" interpretation, but other interpretations would work as well) that give rise to the broad peak. In the second paragraph in the Discussion section, we write:

Under our interpretation that the broad peak arises from FOs, these FOs locally have the same (broken) translational symmetry as the CDWs, so they serve as a conjugate field that induces a crossover behavior of the CDWs above the genuine transition temperature, which we believe is T_{LO} .

Importantly, then in the next paragraph, we discuss another important aspect of the sharp peak's temperature dependence:

But as can be seen from Fig. 2(d), the induced CDW crossover is rather weak in the sense that the majority of the sharp peak's intensity forms below T_{LO} .

It is exactly the fact that the sharp peak has its own transition temperature T_{LO} that has motivated us to write at the end of the second-to-the-last paragraph of our revised manuscript:

T_{LO} can thus be regarded as the "purely electronic" CDW ordering temperature, around which the domains spontaneously merge together, making mobile charge domain walls most noticeable.

In other words, while we agree with the Reviewer that the sharp peak is initially "secondary" to the broad peak, in the sense that the broad peak sets the stage for the sharp peak to develop, it is actually the sharp peak that dominates at the end in the zero-temperature limit. This is fully consistent with our understanding that the sharp peak represents the primary CDWs, whereas the broad peak represents a variation of the CDW correlations (or even simply FOs) that manages to develop from a slightly higher temperature under the help of disorder.

"Hence, although I support the publication of this article in Nature Communications, I recommend that the editor makes sure that the distinction between the experimental result and

the offered interpretation is made clear in the title, abstract, introduction, and conclusion of the article. Many readers focus on these parts only and it is important not to deceive them by letting them think that the article proves without any doubt that FO and CDW coexist in ZrTe₃. For example, "Distinct fingerprints" could be changed to "Possible indications", "These observations demonstrate" to "We interpret these observations", and so on so forth."

We thank the Reviewer for raising these valuable suggestions. The Reviewer's prediction about most readers' behavior probably contains a grain of true. Accordingly, we have thoroughly revised the wording in our manuscript to make it clear that the interpretation about FOs is not uniquely proven by the experimental data. Instead, "FOs + CDWs" represents our "most likely" way of looking at our experimental results altogether.

Our revisions include changing the manuscript title, abstract, and many of the discussion paragraphs. The easiest way to keep track of the changes is to look at the auxiliary file that we provide along with our resubmission, where all major modifications are indicated in blue fonts. With these revisions, the main messages of our manuscript stated in the revised title and abstract would remain valid even if the FO part of the interpretation is replaced by an alternative scenario. We are quite happy with the revised emphasis in these parts and hope the Reviewer would consider it a significant improvement of the manuscript.

Response to Reviewer #3

(Remarks to the Author):

“I still like the paper – the data is noteworthy on two fronts. The two peaks in the diffraction plots and the time-dependent speckles are both observations that are worthy of publication in *Nature Communications*. However, I still have some reservations about the paper in its current form and with some aspects of the interpretation.”

We thank the Reviewer for the time and effort examining our manuscript again, and are pleased by the Reviewer’s favorable recommendation for publication in *Nature Communications*. In the following we address the Reviewer’s remaining reservations.

“Before I discuss that, however, I should mention that parts of the manuscript are greatly improved. The authors have included length scales in the paper, which has greatly improved its accessibility and the general understanding of the experimental phenomenology. Furthermore, the discussion of the two peaks has been improved with the “broad” and “sharp” peak terminology.”

We appreciate the Reviewer’s positive feedback on our previous revision, which we had indeed spent a lot of effort to do. In the case of our work, we find it particularly helpful to “decouple” experimental facts and basic reading of the results from the more in-depth interpretations – we and others may not agree on every interpretation and assessment we make, but it is important for everyone to see the facts in the data and be able to appreciate why the others might be thinking differently. We try to do this again in our present revision.

“Moving onto the interpretation of the experimental results – I believe that there are still substantial problems that need to be resolved in this regard. It is possible that I am partially at fault for this, as upon re-reading my previous review, it seems like what I said may not have been completely clear to the authors. If this was the case, I do apologize to the authors.

In my previous review, I was not suggesting that the FOs and short-ranged CDWs pinned by impurities were the same phenomenon or difficult to tell apart necessarily. I believed that the authors were conflating the two physically different concepts.”

We might have indeed somewhat misunderstood the Reviewer from the previous correspondence, but the Reviewer’s points back then had nevertheless motivated us to improve

the manuscript and are well-appreciated. As a matter of fact, we did not mix up FOs and short-ranged CDWs from the first place, as the two are well-known to be distinct phenomena, at least conceptually. And as the Reviewer correctly stated at a later point in his/her remarks this time, FOs are not expected to have strong temperature dependence, so it is clear that we cannot attribute any of our temperature-dependent signals to *conventional* FOs that are present in all metals.

As we explain in detail below when responding to the Reviewer's itemized remarks, we are in fact in a dilemma of how to call things. Regarding the physical origin of the broad peak, which is now at the focus of our discussion with all our reviewers, we can either attribute it to an *unconventional* type of FOs that receive positive feedback from a deformable lattice and hence become self-amplified, as we have done in our manuscript, or to "CDWs pinned by impurities" as suggested by the Reviewer.

The difficulty with the former way is that the FOs in our scenario are actually different from those that most people are familiar with, and therefore we had gone into lengths to explain what we precisely meant, already in our previous revision. The difficulty with the latter way, in contrast, is in the physics: We already have another type of "pinned CDWs" in our sample, namely the ones that produce the sharp diffraction peak along with static speckles at $T_{LO} < T < T_{CDW}$, so the "pinned CDWs" interpretation for the broad peak would require *two* types of coexisting pinned CDWs that actually behave differently. As have been suggested by another reviewer, it is not impossible that pinned CDWs very close to and at distance from disorder could indeed behave differently, and we agree with all the reviewers that the "pinned CDWs" alternative scenario deserves to be added in the discussion. Yet still, the "pinned CDWs" scenario requires additional understanding of the pertinent parameters' temperature dependence, in order for us to decide whether it works.

Therefore, with respect, we wish to maintain our original interpretation based on FOs as the "most likely" scenario, and discuss the alternatives in a fashion that is as fair as possible. We have accordingly changed the wording in the title, abstract, and many discussion paragraphs so that the main message of our manuscript does not depend on the particular choice of this part of the interpretation.

We have noticed that the word "pin" might also produce some ambiguity, since we are dealing with both (fast) CDW fluctuations and (slow) mobile CDW domain walls. Therefore along with our present revision, we are using words such as "stabilize", "nucleate", and "induce", to refer to crossover-like behavior of the CDWs caused by the presence of disorder, and the word "pin" exclusively in the context of mesoscopic domain pinning.

“The question I wished to raise in the previous review ultimately boils down to the following: Is the observation of the “broad” peak clearly the result of FOs? With the data presented, I cannot convince myself that this is the case. It seems to me more likely that the observations seen here should be attributed to short-ranged CDWs pinned by impurities. If the authors do not think that this is the case, I think it is necessary that they clarify why, which I don’t think they have done adequately in the current manuscript or in the response to my previous review.”

(And from a later place of the Reviewer’s remarks)

“Ultimately, what I am trying to say is that one can probably attribute most of the observations here to CDW fluctuations that get pinned by impurities. For instance, it wouldn’t be surprising for these CDW fluctuations to get pinned by impurities close to, but above T_{LO} . Even below T_{LO} , near the impurity sites, the CDW texture can still get warped leading to a slightly different wavevector in the vicinity of the impurities. As the CDW becomes more rigid with decreasing temperature, the “broad” peak would thus weaken. This would also explain why the peak persists below T_{LO} .”

We agree with the Reviewer on the existence of this possible alternative interpretation of the broad peak. As we mentioned above, this scenario essentially requires the presence of *two* types of “pinned” CDWs for $T_{LO} < T < T_{CDW}$ (whereas below T_{LO} , we could perhaps say that one of them becomes fully spontaneous while the other becomes distorted by the disorder), and they need to behave differently. This is possible, if the disorder somehow manage to influence the CDWs nearby, in addition to merely providing the pinning potential. However, one then needs to add to the scenario why the additional influence of disorder only gets to gradually develop below T_{CDW} , rather than being present at all temperatures. In our opinion, because of this missing link in the “pinned CDWs” interpretation, our interpretation based on FOs is somewhat more natural and simpler in logic.

In our revision, we have added a new paragraph in the Discussion section, which reads:

There are alternative, albeit in our opinion more remote, ways to understand the origin of the broad peak and its momentum detuning from the sharp peak. One possibility is that it arises from *directly* induced CDWs in the immediate neighborhood of disorder sites. In contrast, in both this scenario and our original interpretation based on FOs, the sharp peak at $T_{LO} < T < T_{CDW}$ arises from CDWs that are *indirectly* induced and pinned at distance from the disorder, since it dominates the signal and produces the static speckles, and because it continuously evolves below T_{LO} into the peak of truly long-range CDWs. Regarding the detuning in \mathbf{q} between the two peaks, although the presence of impurity in $ZrTe_3$ may locally modify the electronic structure [40] or create topological defects in the CDWs, it remains to be investigated why the detuning effect gradually develops below T_{CDW} and is not present at T_{CDW} .

This paragraph is followed by the paragraph in which thermal diffuse scattering is discussed. Since dynamic CDWs (which may arguably boil down to phonon modes) are at present actively discussed in the community of high- T_c cuprates (see, *e.g.*, *Science* **365**, 906 (2019)), we wish to discuss it as an alternative scenario on equal footing as the pinned CDWs scenario.

“Additionally, there are a couple observations which go against their interpretation based on FOs.

First, as raised by one of the other reviewers in the last round, the fact that the peaks show up at the same temperature seems contrary to what one would expect of FOs. In Ref. 41, Pouget et. al. claim to see a temperature-independent signal due to FOs.”

The temperature dependence of FOs is a subtle issue. In scattering experiments, *i.e.*, by looking at diffraction-peak amplitudes, electrons’ temperature-dependent self-energy near the Fermi level should in fact affect the prominence of signals due to FOs. This expectation lies in the theoretical analysis of our Refs. [18-21], and it has been suggested to hold true by the experimental study of our Ref. [42]. The sample studied in the article referred to by the Reviewer is heavily doped and disordered, and it is no surprise that scattering signals due to FOs exhibit little change up to relatively high temperatures, because the quasiparticles’ lifetime is limited by impurity rather than thermal scattering from the first place.

In our response to an earlier remark by the Reviewer, we have attempted to clarify that our FO interpretation is based on a special type of FOs that undergo a self-amplification due to mutual positive feedback between the electrons and a deformable lattice. It is this self-amplification below T_{CDW} that gives rise to the strong temperature dependence of the broad peak’s intensity, not the generic property of conventional FOs.

We feel that the above points are already mentioned to some extent in a scattered fashion here and there in our manuscript. Thus, while we agree with the Reviewer that our FO interpretation and nomenclature might raise questions in readers’ mind initially, those questions should get resolved as they read on.

“Second, as the authors themselves state in the manuscript, the FOs should result in the observation of a contour and not a sharp peak in reciprocal space. I don’t fully understand their explanation in the text as to why this wouldn’t be the case.”

Since the FOs that we are referring to are those that can be enhanced by electron-phonon coupling, their momentum-space profile is expected to carry the fingerprint of such coupling matrix elements. In particular, in our earlier study (Ref. [34]) we have demonstrated that the electron-phonon coupling in $ZrTe_3$ is rather momentum-selective. We have revised the sentence that explains why the FOs can manifest themselves as a momentum peak, which now reads:

The strong momentum (\mathbf{k} and \mathbf{q}) dependence of such feedback mechanism [34] may explain why the FO signals materialize in the form of a reciprocal-space peak, rather than a contour [17], as Kohn anomalies which require electron-phonon coupling also commonly exist only in highly restricted momentum regions [6,38].

We hope that the above revision helps to clarify the reason that the Reviewer was asking for.

“I have a two minor comments which the authors might want to take into consideration:”

“1) The paragraph on pg. 4 that begins “We believe that the discrepancy is caused by...” claims that the lattice somehow undergoes long-range ordering at a higher temperature than the electrons (long-range order is clearly observed at 63K in the paper by Hoesch). This claim is highly dubious – it is electron-phonon coupling that is driving the transition in the first place.”

It is not our intention to suggest that the lattice undergoes long-range ordering before the electrons, and we agree with the Reviewer that the two are coupled and must share the same entrance into a long-range ordered state. In the work of Hoesch *et al.*, Fig. 4(a), if one looks closely, it can be found that at 63 K the momentum width of their peak is still decreasing with cooling.

What we intended to explain as “the discrepancy” is about the *intensity* rise below 63 K. It is of no doubt that with hard X-rays, Hoesch *et al.* demonstrated that the lattice starts to deform strongly below 63 K; and now with resonant X-ray scattering, we are in a position to clarify which electrons are responding to (or causing) such deformation, since RXS is mainly sensitive to electron-density modulations. It turns out that we see a corresponding rapid increase below 63 K only in the broad peak’s intensity, but not in the sharp peak’s, and this has been the main purpose of the paragraph specified by the Reviewer. We feel that there is no problem with the narrations in this paragraph in the context of the preceding paragraph.

“2) The paragraph on pg.4 that begins “A possible alternative explanation...” is not necessary. It seems obvious from the temperature and wave vector dependence that the peak does not arise due to thermal diffuse scattering.”

We thank the Reviewer for this comment. After posting our preprint on the arXiv, some readers have contacted us, raising questions related to thermal diffuse scattering. As we have mentioned earlier, dynamic CDWs are at present actively discussed in the community of high- T_c cuprates, and we wish to keep the discussion. But to avoid an interruption in the sub-section where the paragraph previously belonged, we have now moved the paragraph into the Discussion section.

“Ultimately, the previous revision did improve the paper in some parts but worsened it in others. The data in the paper is great and deserves publication, but the authors need another revision to sort out a clear and consistent interpretation of their results. From what I gather, the authors somehow see the FOs as related to the CDW formation. I regard FOs and CDWs pinned by

impurities as two distinct physical phenomena. If the authors are convinced that what they are seeing come from FOs, they must explain why with more clarity and more convincingly.”

We thank the Reviewer again for the favorable assessment of our work and the valuable suggestions. With our revision and our response to the Reviewer above, we hope that we have addressed thoroughly the Reviewer’s remaining reservations.